# MLL1 is required for maintenance of intestinal stem cells

**Neha Goveas**[1], **Claudia Waskow**[2¤a¤b], **Kathrin Arndt**[2], **Julian Heuberger**[3¤c], **Qinyu Zhang**[1], **Dimitra Alexopoulou**[4,5,6], **Andreas Dahl**[4], **Walter Birchmeier**[3], **Konstantinos Anastassiadis**[7], **A. Francis Stewart**[1,8]*, **Andrea Kranz**[1]*

**1** Genomics, Center for Molecular and Cellular Bioengineering, Biotechnology Center, Technische Universität Dresden, Dresden, Germany, **2** Institute for Immunology and Department of Medicine III, Technische Universität Dresden, Dresden, Germany, **3** Laboratory of Signal Transduction in Development and Cancer, Max Delbrück Center for Molecular Medicine, Berlin, Germany, **4** DRESDEN-concept Genome Center, Center for Molecular and Cellular Bioengineering, Technische Universität Dresden, Dresden, Germany, **5** Paul Langerhans Institute Dresden of Helmholtz Zentrum München, University Hospital Faculty of Medicine Carl Gustav Carus, Technische Universität Dresden, Dresden, Germany, **6** German Center for Diabetes Research, Neuherberg, Germany, **7** Stem Cell Engineering, Center for Molecular and Cellular Bioengineering, Biotechnology Center, Technische Universität Dresden, Dresden, Germany, **8** Max Planck Institute of Molecular Cell Biology and Genetics, Dresden, Germany

¤a Current address: Regeneration in Hematopoiesis, Leibniz-Institute on Aging, Fritz-Lipmann-Institute (FLI), Jena, Germany
¤b Current address: Institute of Biochemistry and Biophysics, Faculty of Biological Sciences, Friedrich-Schiller-Universität Jena, Jena, Germany
¤c Current address: Department of Hepatology and Gastroenterology, Charité University Medicine, Berlin, Germany
* francis.stewart@tu-dresden.de (AFS); andrea.kranz@tu-dresden.de (AK)

**Data Availability Statement:** All relevant data are within the manuscript and its Supporting Information files. RNA sequencing data have been deposited in the Gene Expression Omnibus under accession number GSE 157285 and GSE 174071.

## Abstract

Epigenetic mechanisms are gatekeepers for the gene expression patterns that establish and maintain cellular identity in mammalian development, stem cells and adult homeostasis. Amongst many epigenetic marks, methylation of histone 3 lysine 4 (H3K4) is one of the most widely conserved and occupies a central position in gene expression. Mixed lineage leukemia 1 (MLL1/KMT2A) is the founding mammalian H3K4 methyltransferase. It was discovered as the causative mutation in early onset leukemia and subsequently found to be required for the establishment of definitive hematopoiesis and the maintenance of adult hematopoietic stem cells. Despite wide expression, the roles of MLL1 in non-hematopoietic tissues remain largely unexplored. To bypass hematopoietic lethality, we used bone marrow transplantation and conditional mutagenesis to discover that the most overt phenotype in adult *Mll1*-mutant mice is intestinal failure. MLL1 is expressed in intestinal stem cells (ISCs) and transit amplifying (TA) cells but not in the villus. Loss of MLL1 is accompanied by loss of ISCs and a differentiation bias towards the secretory lineage with increased numbers and enlargement of goblet cells. Expression profiling of sorted ISCs revealed that MLL1 is required to promote expression of several definitive intestinal transcription factors including *Pitx1*, *Pitx2*, *Foxa1*, *Gata4*, *Zfp503* and *Onecut2*, as well as the H3K27me3 binder, *Bahcc1*. These results were recapitulated using conditional mutagenesis in intestinal organoids. The stem cell niche in the crypt includes ISCs in close association with Paneth cells. Loss of MLL1 from ISCs promoted transcriptional changes in Paneth cells involving metabolic and

**Funding:** This work was supported by the Deutsche Forschungsgemeinschaft (DFG) STE903/7-3, STE903/12-1, STE903/13-1 (AFS) and KR2154/6-1 (AK). The work was supported by grants from the DFG WA2837/3-2, WA2837/6-1, WA2837/7-1, and the SAW program on collaborative excellence of the Leibniz Association (K243/2019) to CW. This work was supported by central resources of the MDC to JH. The funders had no role in study design, data collection and analysis, decision to publish, or preparation of the manuscript.

**Competing interests:** The authors have declared that no competing interests exist.

stress responses. Here we add ISCs to the MLL1 repertoire and observe that all known functions of MLL1 relate to the properties of somatic stem cells, thereby highlighting the suggestion that MLL1 is a master somatic stem cell regulator.

## Author summary

The ability of stem cells to replenish cellular lineages is critical for the maintenance and integrity of postnatal life. In mammals, the blood and the intestinal lining are two of the highest turnover tissues and both are anchored by stem cells. For blood, hematopoeitic stem cells require MLL1 (mixed lineage leukemia), which is an epigenetic regulator that methylates histone 3 on lysine 4 (H3K4). MLL1 was discovered because it is mutated in about 10% of all human leukemias prominently including early onset childhood leukemias. In this study we show that MLL1 is also required for maintenance of intestinal stem cells (ISCs). MLL1 is expressed in the intestinal crypt, which harbors the ISC niche and transit amplifying cells but is not expressed in the intestinal villi that line the gut. Conditional mutagenesis of *Mll1* in adult mice rapidly leads to intestinal failure, wasting and death. ISCs are lost and the niche is perturbed. Loss of ISC identity can be recapitulated by conditional mutagenesis of *Mll1* in intestinal organoids. In ISCs, MLL1 is required to maintain the expression of several DNA binding transcription factors that are implicated in regulating ISC cellular identity. Hence we report that two mammalian tissues with very high turnover depend upon MLL1 to maintain their respective stem cells. Together with other indications, MLL1 appears to be specifically related to adult stem cell properties.

## Introduction

Stem cells are cornerstones of tissue biology, ensuring homeostasis and regeneration in many organs, including epithelial tissues such as skin, intestine and mammary gland [1]. Stem cells are characterized by multipotency, which is the ability to differentiate into a restricted number of defined cell types, and self-renewal, which is the capacity to undergo infinite replicative cycles without losing stem cell identity [2]. The remarkable capacities of stem cells, particularly the restricted specificities of multipotency, rely on interplays between specific transcription factors and distinct epigenetic landscapes. Whereas many transcription factors involved in stem cell maintenance and differentiation have been defined, epigenetic contributions are proving more elusive. For example, the transcription factor hierarchies in the stem cell paradigm, hematopoiesis, have been elegantly dissected [3]. However the contributions of DNA and histone methyltransferases to hematopoiesis are still emerging and indicate both specificities and the deeper complexities of epigenetic regulation [4–8].

Methylation of histone 3 on lysine 4 (H3K4) is one of the most conserved and widespread epigenetic systems [9]. H3K4 is methylated in euchromatic regions, with trimethylated H3K4 (H3K4me3) on nucleosomes surrounding active promoters, H3K4me2 marking transcribed regions and H3K4me1 relating to enhancers and active chromatin in general [10–14]. Mammals have six Set1/Trithorax-related H3K4 methyltransferases (H3K4MTs) that are encoded by three pairs of paralogous sister genes namely, *Mll1* (*Kmt2a*) and *Mll2* (*Kmt2b*), *Mll3* (*Kmt2c*) and *Mll4* (*Kmt2d*), *Setd1a* (*Kmt2f*) and *Setd1b* (*Kmt2g*). Each of the six H3K4MTs resides in their own, large, protein complex. However all six complexes are based on a four membered scaffold termed WRAD for the subunits WDR5, RBBP5, ASH2L and DPY30 [15]

or sometimes called COMPASS [9]. Functional differences between the six complexes potentially arise from the presence of additional subunits, which are usually shared by paralogous pairs or sometimes uniquely found in one of the six complexes. For example, MENIN is a subunit of both MLL1 and 2 complexes [16] but so far, no subunit unique to either paralogous complex has been described.

Mixed lineage leukemia (*MLL1*) was the first mammalian gene identified as a Trithorax homologue and subsequently found to encode a mammalian Set1/Trithorax-type H3K4MT [17,18]. In mice, MLL1 is first required at embryonic day 12.5 (E12.5) for definitive hematopoiesis [19,20] and also for the maintenance of adult hematopoietic stem cells (HSCs) [21,22]. *MLL1*, but not its paralogue, *MLL2*, is a proto-oncogene because it is activated by chromosomal translocations to promote leukemias, notably without additional mutagenesis [23,24]. Over 80 translocation partners have been identified including AF6 and AF9 [25]. Mouse studies indicated that *MLL1-AF6* and *-AF9* leukemias rely on MLL2 expression [6] and *MLL1-AF9* leukemiogenesis is entirely conveyed by overexpression of HOXA9 [26]. Conditional mutagenesis has also revealed MLL1 functions in satellite cells [27] and postnatal neural stem cells (NSCs) [28]. These observations raise the possibility that MLL1 regulates specific functions in stem cell compartments.

Due to the high turnover and hierarchical architecture of the intestinal epithelium, intestinal stem cells (ISCs) have become an adult stem cell paradigm. ISCs have been identified as either actively cycling crypt base columnar cells (CBCs) or quiescent label-retaining cells (LRCs) located at the +4 position from the crypt base [29]. LGR5 (Leucine-rich repeat-containing G protein-coupled receptor 5) is the characteristic marker for the CBC class of ISCs [30], which generates the transit amplifying (TA) daughter cells. Proliferation of the TA cells generates the cells that line the villi: absorptive enterocytes, secretory goblet, enteroendocrine; as well as Paneth cells [31]. Except for Paneth cells, these cell types take 3 to 5 days to migrate out of the crypt up the villi, where they are shed into the intestinal lumen. Paneth cells stay in the crypt and interweave with ISCs to form the intestinal stem cell niche that is sustained by Wnt signaling [31,32]. Quiescent LRCs serve as a stem cell reserve to replace damaged ISCs and are active only during stress or injury [33]. Enterocytes, preterminal enteroendocrine cells, goblet cell precursors and Dll1⁺ secretory progenitors are also notable for their plasticity and can serve as a reservoir for lost stem cells [34–36]. Mature Paneth cells also show an injury-activated conversion to a stem cell like state [37]. Together with new perceptions in hematopoiesis [4,6,38], the dynamic plasticity of the intestinal crypt has expanded the stem cell paradigm [39], especially during replenishment after damage or inflammation.

To examine whether MLL1 plays non-hematopoietic roles in the adult, we employed ligand-induced conditional mutagenesis [40,41] using a tamoxifen-inducible *Rosa26-CreERT2* line (*RC*) [42] for near-ubiquitous Cre recombination to discover that MLL1 is required for the maintenance of the ISC compartment and the balance between secretory and absorptive cell lineages in the adult intestine.

## Results

### Intestinal functions collapse after loss of MLL1 in adult mice

To explore MLL1 functions, we utilized a multipurpose allele that can be converted from one state to another using FLP and Cre recombination (S1A–S1C Fig) [43,44]. Homozygous embryos carrying the targeted gene trap allele, *Mll1^{A/A}*, developed normally until E12.5 when they displayed pallor of the liver and were smaller (S1D Fig). After E13 no live *Mll1^{A/A}* embryos were found (S1 Table). After FLP and Cre recombination, *Mll1^{FC/FC}* embryos carrying the frame-shifted allele displayed the same phenotype indicating that both A and FC are

true null alleles as expected from the loss of MLL1 protein [45]. Furthermore the $Mll1^{A/A}$ and $Mll1^{FC/FC}$ phenotype recapitulated another likely null allele [6]. Homozygous mice from these three mutant alleles present the same embryonic lethality due to the failure to engage definitive hematopoiesis [21]. Consequently we suggest that the diverse early developmental phenotypes reported previously for mutations in $Mll1$ [46,47] were likely due to dominant negative $Mll1$ alleles.

To identify postnatal roles of MLL1, ligand-induced conditional mutagenesis using $Rosa26$-$CreERT2$ ($RC$) was applied to two month old adults. Mice lacking MLL1 developed severe bone marrow cytopenia and died or had to be sacrificed on average within two weeks (Fig 1A and 1B). As previously reported using the same conditional allele, $in\ vitro$ deletion of $Mll1$ in KSL-enriched HSCs from $Mll1^{F/F;\ RC/+}$ mice resulted in significant downregulation of $Hoxa9$, $Meis1$, $Mecom/Evi1$ and $Prdm16$ [6].

To bypass the bone marrow related lethality and thereby uncover non-hematopoietic phenotypes, bone marrow from wild type (wt) B6.SJL mice was transplanted into lethally irradiated $Mll1^{F/+;\ RC/+}$ or $Mll1^{F/F;\ RC/+}$ mice. After stable engraftment tamoxifen gavage induced widespread Cre-mediated excision of $Mll1$ (Fig 1C). Examination of the bone marrow confirmed the successful and near-complete reconstitution of the hematopoietic stem cell compartment by wt donor cells. FACS analysis for KSL-Slam enriched HSCs showed comparable frequencies in bone marrow transplanted (BMTx) $Mll1^{FC/+;\ RC/+}$ and $Mll1^{FC/FC;\ RC/+}$ mice with the hematopoietic compartment comprised only of wt cells of CD45.1 origin (Fig 1C). Notably, the BMTx $Mll1^{FC/FC;\ RC/+}$ mice suffered from diarrhea and wasting but heterozygous BMTx $Mll1^{FC/+;\ RC/+}$ did not (Fig 1D). These data indicate that MLL1 is not only required in the hematopoietic compartment but is similarly critical elsewhere.

The small and large intestine harbor stem cell compartments at the base of the intestinal crypts. In the crypt, MLL1 is expressed in ISCs and the TA compartment, whereas it is absent in differentiated cells above the TA compartment (Figs 1E and S1E) [48]. RNA profiling of sorted ISCs and Paneth cells confirmed expression of $Mll1$ and the other five H3K4MTs in both cell types however all are more strongly expressed in ISCs (S1F Fig). In the mutant small intestine of BMTx $Mll1^{FC/FC;\ RC/+}$ mice, expression of MLL1 was efficiently ablated and ISC markers OLFM4 (olfactomedin 4) and SOX9 were lost (Fig 2A). Concordantly, proliferation in the crypt was reduced as revealed by strongly reduced expression of the mitotic marker Ki67 (Fig 2A). However, no apparent changes in global H3K4 mono-, di- and trimethylation in the intestinal epithelium of $Mll1^{FC/FC;\ RC/+}$ mice were observed (S1G Fig).

Shortened villi with distorted morphology including vacuolar structures at the tip indicated diminished replenishment of cells into the villus (Fig 2B). Furthermore we observed increased numbers of enlarged goblet cells distributed irregularly along the villus and also ectopically in the crypt (Fig 2B). However, Paneth cells appeared unchanged possibly due to their longer life span (Fig 2C). Similarly, as evaluated by chromogranin A and alkaline phosphatase, the enteroendocrine and absorptive lineages appeared to be unaffected (Fig 2C).

Without bone marrow rescue, tamoxifen-induced conditional mutagenesis of $Mll1$ ($Mll1^{FC/FC;\ RC/+}$ mice) showed the same defects with depletion of ISCs, decreased proliferation and a distortion of the secretory lineage (S2A–S2D Fig). Differentiation into the enteroendocrine and absorptive lineage was also apparently unaffected (S2E Fig). Notably the hallmark of Wnt signaling, nuclear β-catenin, was also unaffected (S2F Fig).

## Intestinal epithelium-specific $Mll1$ conditional mutagenesis recapitulates ubiquitous deletion

To delete $Mll1$ exclusively in the adult intestine we employed the tamoxifen-inducible gut epithelium-specific $Villin$-$CreER^{T2}$ line ($Vil$-$Cre$-$ERT2$) [49]. After tamoxifen administration

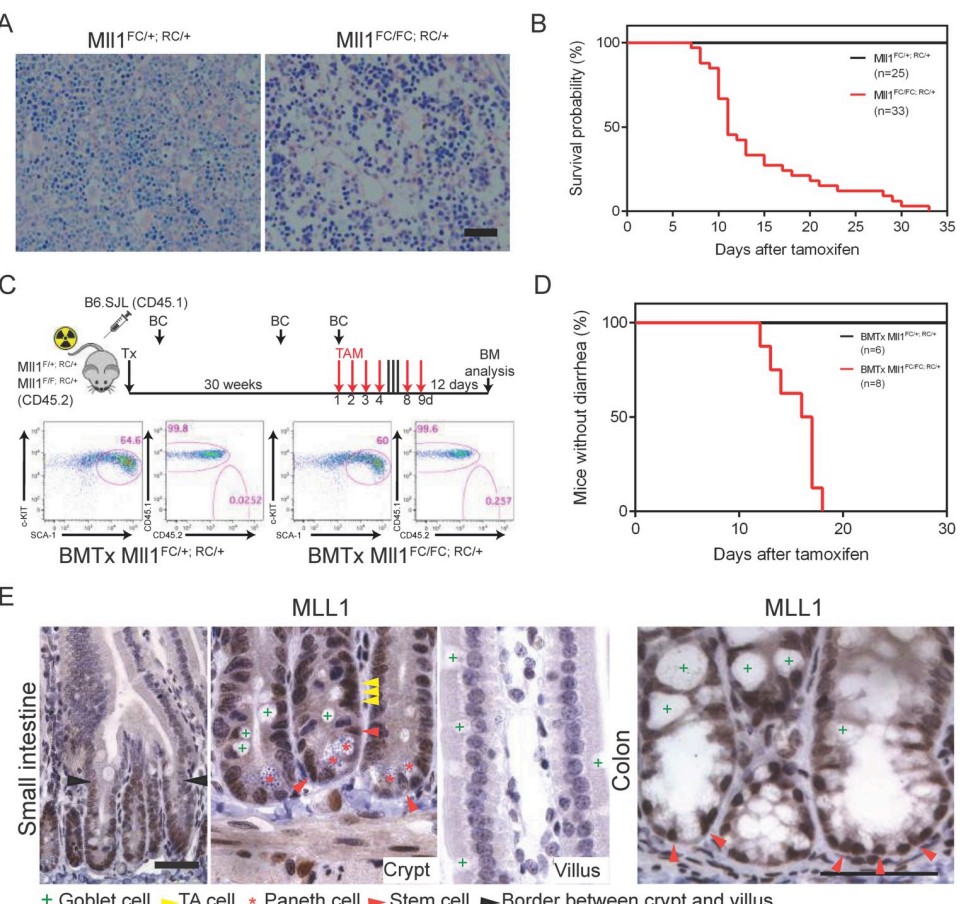

**Fig 1. Loss of MLL1 in adult mice.** (A) Giemsa-stained sections from the femur of $Mll1^{FC/+; RC/+}$ and $Mll1^{FC/FC; RC/+}$ mice. Two weeks after the last tamoxifen gavage mice were sacrificed and the femurs were dissected, decalcified, sectioned and stained. Bone marrow cellularity was severely decreased in $Mll1^{FC/FC; RC/+}$ mice. Scale bar 100 μm. (B) Kaplan-Meier survival curve. The first day of tamoxifen gavage was day zero. All $Mll1^{FC/+; RC/+}$ mice (n = 25) survived whereas all $Mll1^{FC/FC; RC/+}$ mice (n = 33) died within 33 days after tamoxifen induction with a median survival of 11 days. (C) Scheme of the experimental setup for bone marrow transplantation. Donor bone marrow from B6.SJL (CD45.1$^+$) mice was transplanted into lethally irradiated $Mll1^{F/+; RC/+}$ and $Mll1^{F/F; RC/+}$ recipients (CD45.2$^+$). Blood chimerism (BC) was measured three times. After 30 weeks $Mll1$ deletion was achieved by administrating tamoxifen (TAM). FACS analysis for KSL-Slam enriched HSCs (Kit$^+$ Sca1$^+$ Lin$^-$ CD48$^-$ CD150$^+$ CD34$^-$ CD135$^-$) showed comparable numbers in BMTx $Mll1^{FC/+; RC/+}$ and $Mll1^{FC/FC; RC/+}$ mice. Dot plots show Lin$^-$ CD48$^-$ CD150$^+$ CD34$^-$ CD135$^-$ gated bone marrow (BM) cells of indicated genotypes resolved for the expression of c-Kit and Sca-1. Donor and host cells are distinguished by surface markers CD45.1 and CD45.2. (D) Kaplan-Meier analysis for the onset of diarrhea. Tamoxifen was given by gavage for 6 days to $Mll1^{F/+; RC/+}$ (n = 6) and $Mll1^{F/F; RC/+}$ (n = 8). The first day of tamoxifen gavage was day zero. While all BMTx mice with the genotype $Mll1^{FC/+; RC/+}$ remained healthy, all BMTx $Mll1^{FC/FC; RC/+}$ mice developed diarrhea with a median of 16.5 days. (E) Antibody staining (brown) showed that MLL1 is expressed in crypts of the small and large intestine but absent in the villus (hematoxylin, purple). Scale bars are 50 μm.

about half (10/18) of the $Mll1^{FC/FC; Vil-Cre-ERT2/+}$ mice lost weight compared to control mice. These mice displayed wasting and were mostly analyzed three to four weeks after tamoxifen induction (Fig 3A and 3B). Compared to $Mll1^{FC/+; Vil-Cre-ERT2/+}$ littermates, OLFM4 and SOX9 expression, and proliferation were markedly decreased, goblet cells were enlarged and increased in number whereas Paneth and enteroendocrine cell numbers were unchanged (Fig 3C and 3D). However the rest (8/18) of the $Mll1^{FC/FC; Vil-Cre-ERT2/+}$ mice did not display wasting and survived (Fig 3A), which is a result we attribute to insufficient Cre recombination by $Vil-Cre-ERT2$ as noted before [49,50], and the regenerative capacities of the intestinal crypt

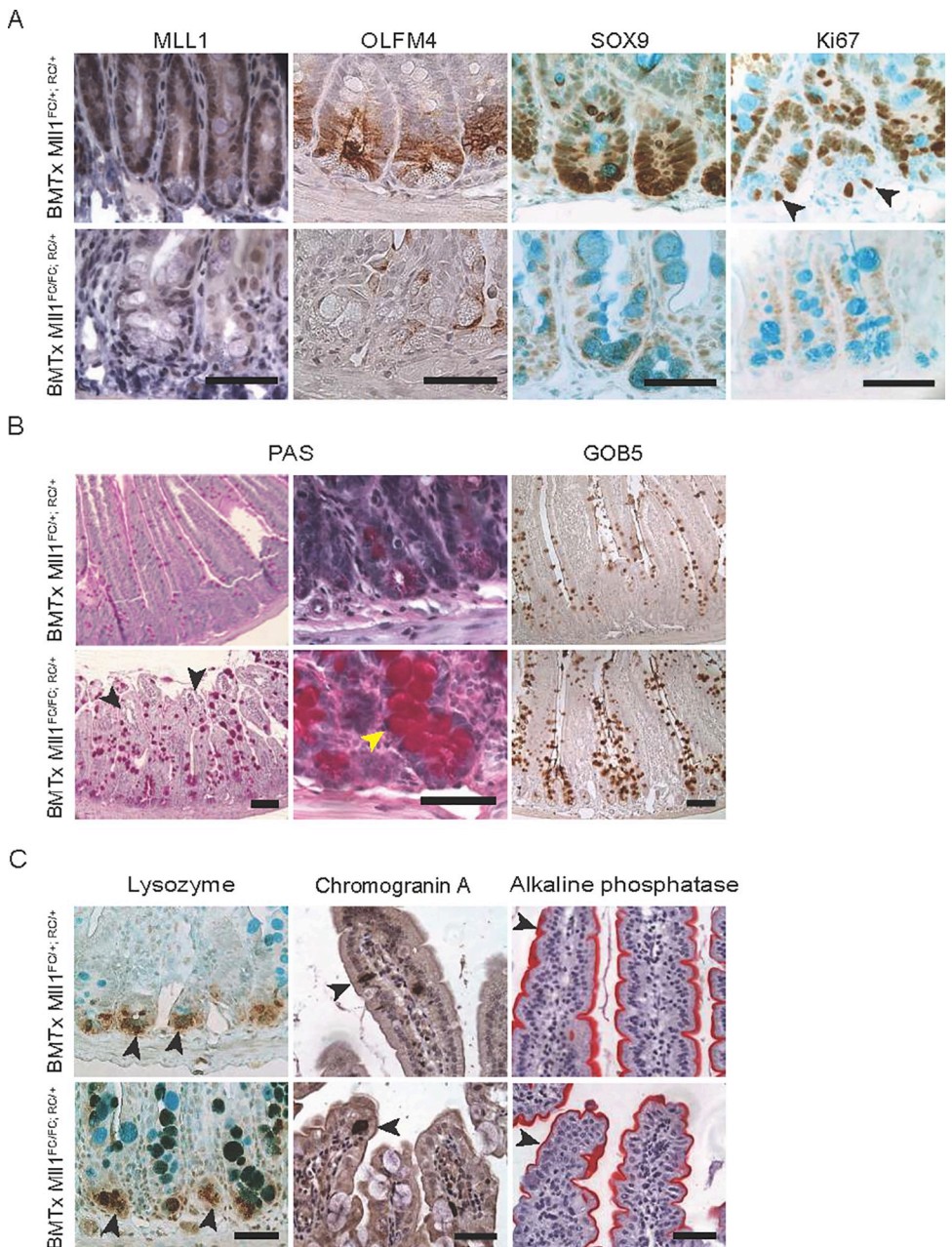

**Fig 2. *Mll1* deletion in BMTx mice leads to loss of stem and proliferating cells and increased goblet cells.** (A) Antibody stainings showed reduced MLL1, OLFM4 and SOX9 expression in BMTx *Mll1^{FC/FC; RC/+}* intestine compared to controls. Hematoxylin was used as a counterstain for MLL1 and OLFM4 immunohistochemistry (IHC). Expression of Ki67, a marker for proliferating cells was reduced in BMTx *Mll1^{FC/FC; RC/+}* mutant intestinal sections. Arrowheads point toward proliferating ISCs. Alcian blue was used as a counterstain for SOX9 and Ki67 IHC, which also revealed enlarged goblet cells (turquoise) in the crypts of BMTx *Mll1^{FC/FC; RC/+}* sections. Scale bars are 50 μm. (B) PAS stain to examine goblet cells in villi and crypts of BMTx intestine. Black arrowheads point towards vacuolar structures. Yellow arrowhead points to a mislocalized goblet cell in BMTx *Mll1^{FC/FC; RC/+}* crypt. Left panels scale bar 100 μm; middle panels scale bar 50 μm. GOB5 antibody staining of BMTx *Mll1^{FC/FC; RC/+}* intestinal sections. Right panels scale bar 100 μm. (C) Left panels; lysozyme antibody staining reveals that Paneth cell numbers remain unchanged. Arrowheads point at Paneth cells. Alcian blue was used as a counterstain and marks goblet cells (turquoise). Middle panels; chromogranin A antibody stain, arrowheads point to the sparse enteroendocrine cells (dark brown) in the villi. Right panels; red enterocytes (arrowheads) covering the villi were visualized by alkaline phosphatase staining. Hematoxylin was used as a counterstain for chromogranin A IHC and alkaline phosphatase histochemical staining. Scale bars are 50 μm.

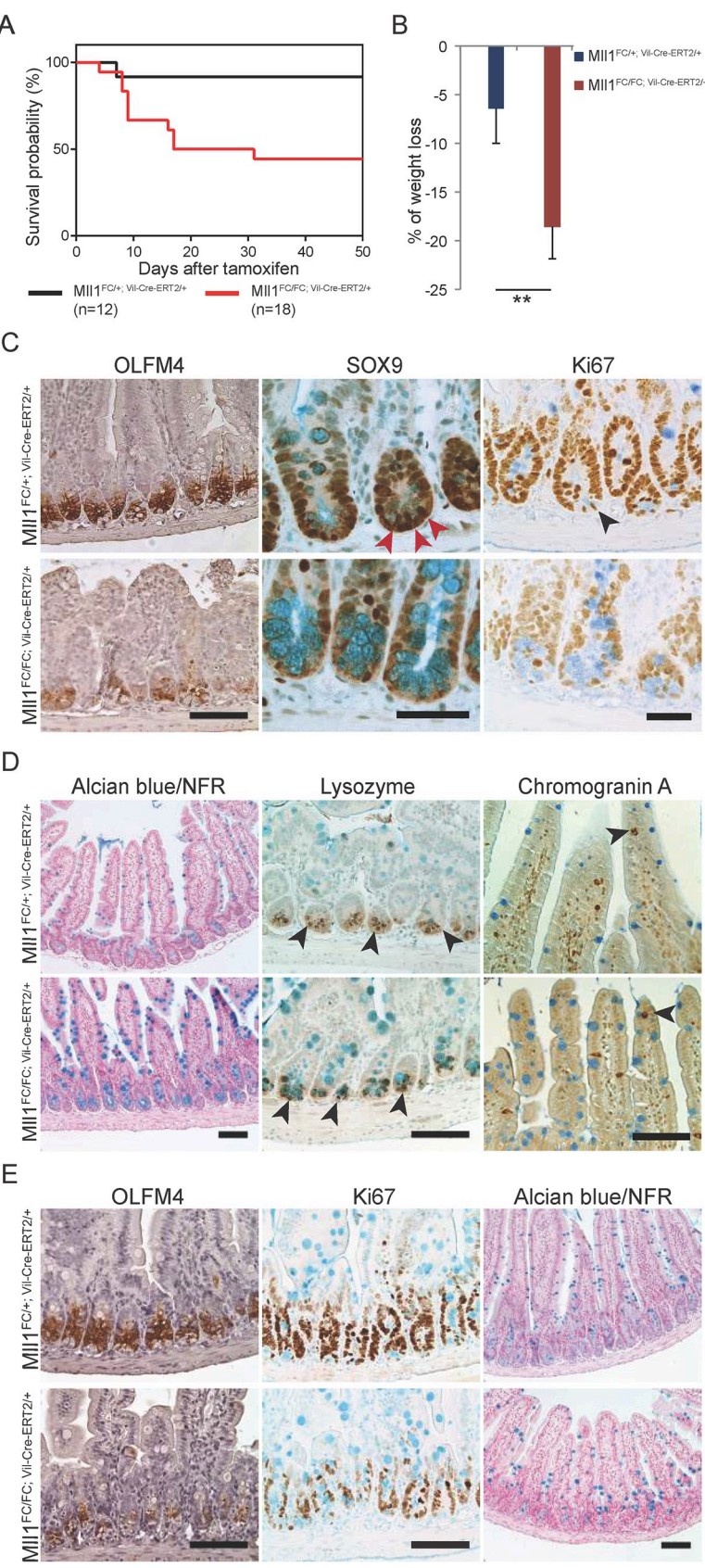

**Fig 3. Decreased ISCs and increased goblet cells after intestinal specific loss of MLL1. (A)** Kaplan-Meier survival curve. Tamoxifen was given by gavage for 6 days to mice with the genotype *Mll1*[FC/+; Vil-Cre-ERT2/+] (n = 12) and *Mll1*[FC/FC; RC/+] (n = 18). The first day of tamoxifen gavage was day zero. The median survival of mutants is 24 days after tamoxifen. **(B)** Percent of weight loss of *Mll1*[FC/+; Vil-Cre-ERT2/+] (n = 6) and of *Mll1*[FC/FC; Vil-Cre-ERT2/+] (n = 10) mice. Mean ± s.d. is shown (p = 0.007, Student's *t*-test). **(C)** Decrease in ISC markers, OLFM4 and SOX9 in *Mll1*[FC/FC; Vil-Cre-ERT2/+] intestinal sections. SOX9[+] ISCs in *Mll1*[FC/+; Vil-Cre-ERT2/+] intestinal sections are marked with red arrowheads. Proliferating cells in the TA compartment as well as proliferative ISCs (arrowhead) are reduced in *Mll1*[FC/FC; Vil-Cre-ERT2/+] sections. Alcian blue was used as a counterstain after staining for SOX9 and Ki67 and marks goblet cells (turquoise). Scale bars are 100 μm for OLFM4 and 50 μm for SOX9 and Ki67. **(D)** Left panels; alcian blue staining of goblet cells with nuclear fast red (NFR) to stain nuclei. Middle panels; Paneth cells visualized by staining of granules containing lysozyme (arrowheads). Right panels; enteroendocrine cells stained with chromogranin A antibody (brown; arrowheads) in the villi. Scale bars are 100 μm. **(E)** Left and middle panels; antibody stainings showed reduced OLFM4 and Ki67 expression in *Mll1*[FC/FC; Vil-Cre-ERT2/+] intestine compared to controls, one week after the first gavage. Hematoxylin was used as a counterstain for OLFM4 and alcian blue was used for Ki67 IHC. Right panel; alcian blue staining of goblet cells with NFR to stain nuclei, one week after the first gavage. Scale bars are 100 μm.

[31]. Concordant with this explanation, some of the surviving *Mll1*[FC/FC; Vil-Cre-ERT2/+] mice sectioned at 4 weeks stained strongly for OLFM4 and Ki67, but also had elevated and enlarged goblet cells (S2G Fig), indicating that Cre deletion had provoked alterations in the crypt during regenerative recovery.

To gather more evidence regarding mechanism and the temporal order of events after loss of MLL1, we analyzed intestinal sections shortly after tamoxifen induction. One to two weeks after the first tamoxifen gavage, the stem cell marker, OLFM4 and the proliferation indicator, Ki67, were markedly decreased. However only a slight increase in the goblet cell population in *Mll1*[FC/FC; Vil-Cre-ERT2/+] intestinal sections was observed (Fig 3E) indicating that changes in ISCs preceded other alterations.

## Transcriptome analysis identifies key intestinal transcription factors as central to the crypt stem cell niche

For transcriptome analysis, ISCs and Paneth cells were sorted using a FACS strategy 4 and 10 days after the final tamoxifen administration from *Mll1*[FC/+; Lgr5-eGFP-CreERT2/+] and *Mll1*[FC/FC; Lgr5-eGFP-CreERT2/+] littermates (S3 Fig). Using 75 base-pair reads, 20–37 million reads per sample with high levels of uniqueness (70–77% in Lgr5[+] ISCs and 60–76% in Paneth cells; S4A–S4C Fig) and comparable mapability (99%) were obtained. Principal component analysis (PCA) revealed that our datasets are in good agreement with published datasets obtained from sorted ISCs and Paneth cells [37,51] (S4D Fig).

We applied DESeq2 to analyze differentially expressed genes (DEGs). In the 4 day Lgr5[+] stem cell profile, only 87 and 49 genes were up- or downregulated after removal of MLL1 at 5% FDR (false discovery rate) (Fig 4A and S1 Data). However, none of the upregulated transcripts were increased by more than 2-fold and by DAVID analysis were mainly related to diverse terms such as 'response to metal ion', 'organic acid metabolic process' and 'regulation of lipid metabolic process' (Fig 4B). In contrast the most significant terms associated with the downregulated mRNAs were 'regulation of gene expression', 'epithelial cell differentiation' and 'cell proliferation' (Fig 4B). The 10 day profile revealed 179 DEGs, of which 105 were upregulated, 74 were downregulated with significant overlaps to the 4 day profile (S1 Data).

Gene set enrichment analysis (GSEA) of both 4 and 10 day ISC profiles revealed that overall ISC signature genes were downregulated whereas goblet cell signature genes were upregulated (Figs 4C, 4D, 4F and S5A and S5B), which concords with our immunohistochemical analyses. We focused on the overlap between the 4 and 10 day downregulated mRNAs. The transcription factors *Pitx2*, *Foxa1*, *Zfp503/Nolz1*, *Pitx1*, *Onecut2* and *Gata4* are prominent (Table 1 and Fig 4E). The expression of *Foxa1* and *Pitx2* was evaluated by qRT-PCR with good agreement

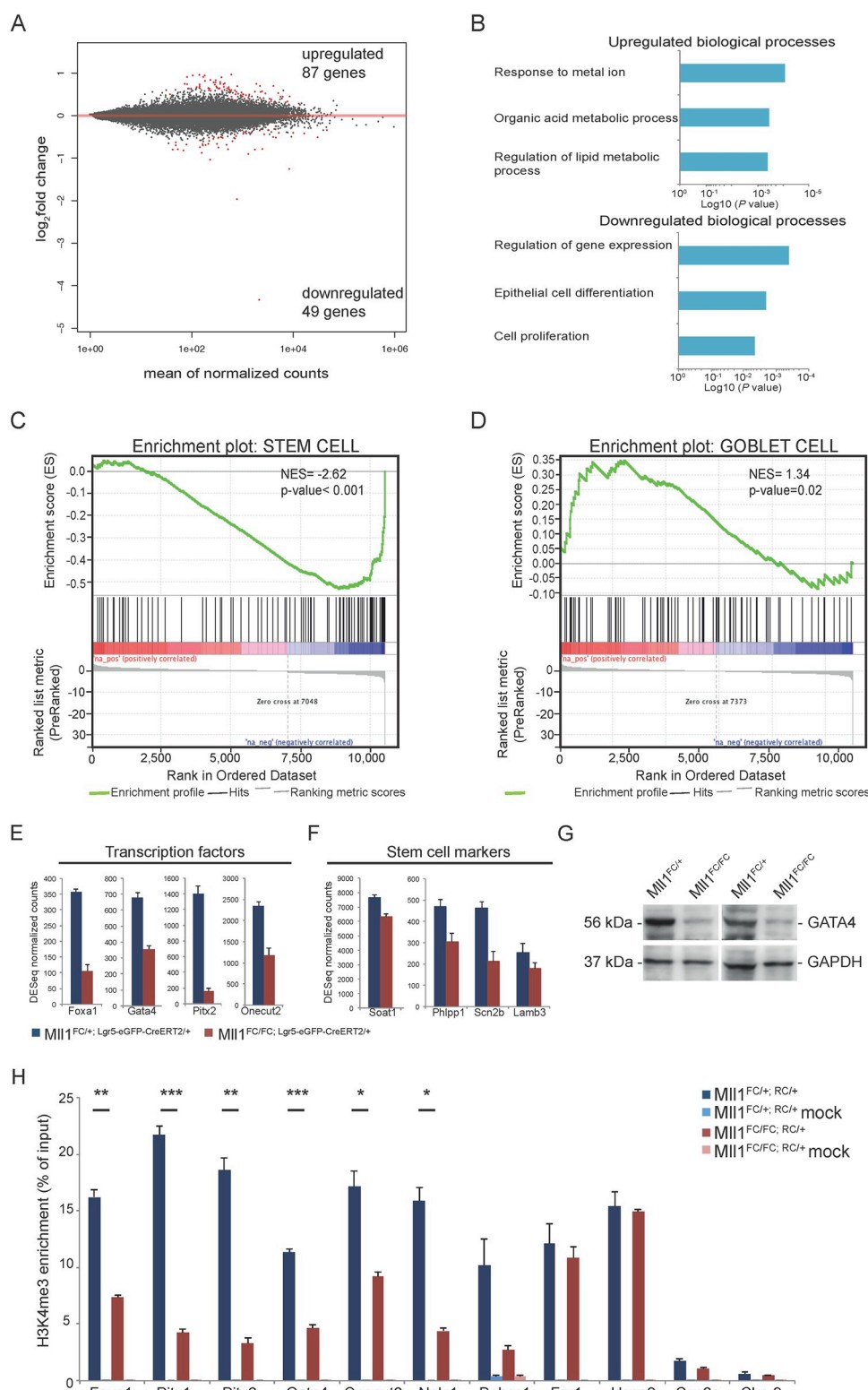

**Fig 4. RNA profiling of *Mll1*^FC/FC; *Lgr5-eGFP-CreERT2/+* and control ISCs.** (A) ISCs were sorted from control (*Mll1*^FC/+; ^*Lgr5-eGFP-CreERT2/+*) (n = 4) and *Mll1*^FC/FC; *Lgr5-eGFP-CreERT2/+* (n = 4) mice 4 days after tamoxifen induction was completed and subjected to RNA profiling. MA plot visualizing the log2-fold change differences according to expression levels. Red dots represent significant DEGs at a 5% FDR. (B) Plots show biological processes (BP) that are enriched in genes up- or downregulated in *Mll1*^FC/FC; *Lgr5-eGFP-CreERT2/+* compared to control ISCs. Analysis was

performed using the gene ontology (GO)/BP/FAT database of DAVID 6.8. (C) (D) GSEA shows significant negative or positive correlation of genes from the stem (C) and goblet cell (D) signature gene set in $Mll1^{FC/FC;\ Lgr5-eGFP-CreERT2/+}$ compared to control ISCs (4 days after tamoxifen). The signature gene sets originate from [87]. NES: normalized enrichment score. (E) DESeq normalized counts for genes coding for transcription factors downregulated in $Mll1^{FC/FC;\ Lgr5-eGFP-CreERT2/+}$ compared to control ISCs (4 days after tamoxifen). Mean+s.d. is shown; n = 4; p<0.05, Wald test. (F) DESeq normalized counts for genes coding for ISC markers downregulated in $Mll1^{FC/FC;\ Lgr5-eGFP-CreERT2/+}$ compared to control ISCs (4 days after tamoxifen). Mean+s.d. is shown; n = 4; p<0.05, Wald test. (G) Western blot analysis for GATA4 protein on crypts from two pairs of $Mll1^{FC/+;\ RC/+}$ and $Mll1^{FC/FC;\ RC/+}$ mice. GAPDH serves as loading control. (H) Representative ChIP-qPCR analysis specific for H3K4me3 occupancy at promoter regions. Crypts from $Mll1^{FC/+;\ RC/+}$ or $Mll1^{FC/FC;\ RC/+}$ mice were crosslinked and chromatin immunoprecipitation was performed with and without (mock) H3K4me3 antibody. Target genes were selected from Table 1. Mean+s.d. is shown; n = 3 *p<0.05, **p<0.01, ***p<0.001, Student's t test.

**Table 1. Common mRNAs downregulated after removal of MLL1 from ISCs and/or organoids.**

| Gene | Description | 4d het | FD | 10d het | FD | P1 het | FD | P3 het | FD |
|---|---|---|---|---|---|---|---|---|---|
| *Pitx2* | Paired like homeodomain 2 | 1399.6 | **8.6** | 854.8 | **24.6** | 609.9 | **1.2*** | 758.4 | **12.1** |
| *Bahcc1* | BAH domain and coiled-coil 1 | 66.4 | **3.8** | 47.7 | **8.1** | 657.7 | **2.4** | 1032 | **42.0** |
| *Foxa1* | Forkhead box A1, *Hnf3α* | 356.8 | **3.4** | 428.4 | **4.6** | 774.2 | **1.7** | 1135.9 | **2.0** |
| *Hspa8* | Heat shock protein A8, *Hsp70C* | 12104 | **2.6** | 7654.5 | **3.0** | 9935.3 | **2.1** | 12432 | **2.4** |
| *Zfp503* | Zinc finger protein 503, *Nolz1* | 113.2 | **2.5** | 166.7 | **7.9** | 683.9 | **2.5** | 775.7 | **2.8*** |
| *Pla2g2a* | Phospholipase A2 group IIA | 291.3 | **2.5** | 215.7 | **3.2** | | | | |
| *Pitx1* | Paired like homeodomain 1 | 71.8 | **2.3** | 75.5 | **6.4** | 1484.7 | **1.7** | 1128.9 | **16.4** |
| *Prss12* | Protease serine 12 neurotrypsin, *Motopsin* | 58.1 | **2.3** | 97.8 | **11.2** | 1054.5 | **1.5** | 938.3 | **4.7** |
| *Scn2b* | Sodium channel subunit 2b | 460.6 | **2.2** | 545.3 | **1.8** | | | | |
| *Cerk* | Ceramide kinase | 198.1 | **2.1** | 141.6 | **1.2*** | 2228.1 | **1.4** | 2918.8 | **2.2** |
| *Onecut2* | Onecut homeobox 2, *Hnf6β* | 2334.5 | **2.0** | 2411.9 | **4.1** | 4344.5 | **1.4*** | | |
| *Casc4* | Cancer susceptible 4 | 120.5 | **2.0** | 255.1 | **2.7** | | | | |
| *Gata4* | GATA binding protein 4 | 676.4 | **1.9** | 508.2 | **3.7** | 3535.0 | **1.4** | 5820.7 | **3.3** |
| *Far1* | SID1 transmembrane family1 | 402.3 | **1.8** | 231.8 | **4.9** | | | | |
| *Fam149a* | Toll-like receptor 3? | 170.7 | **1.8** | 193.9 | **2.1** | 435.3 | **1.2*** | 659.2 | **2.0** |
| *Klhl23* | Kelch-like 23 | 383.8 | **1.7** | 291.5 | **1.4*** | 452.7 | **1.7** | 341 | **1.4*** |
| *Lmcd1* | LIM & cysteine rich domains 1 | 432.8 | **1.2** | 263.6 | **2.1** | 210 | **1.6** | 248.2 | **2.2** |
| *Cdkn1b* | Cyclin-dependent kinase inhibitor 1B | 330.8 | **1.6** | 482.3 | **1.6** | 1565.8 | **1.5** | 2882.6 | **1.8** |
| *Ces2g* | Carboxyl esterase 2 | 275.2 | **1.6** | 209.9 | **3.6** | | | 4055.9 | **2.8** |
| *Phlpp1* | PH domain leucine rich phosphatase | 468.7 | **1.6** | 480.5 | **2.1** | 481.6 | **1.5** | 932.0 | **1.8** |
| *Zfpm1* | *FOG1*, Friend of Gata 1 | 198.5 | **1.4** | 216.6 | **2.0** | 3756.9 | **1.2** | 5252.1 | **1.7** |
| *Lcp1* | Lymphocyte cytosolic P1 | 824.4 | **1.4** | 662.0 | **2.3** | 2904.3 | **1.1*** | 5092.5 | **2.6** |
| *Gdf9* | Growth differentiation factor 9 | | | | | 190.7 | **2.0** | 771.3 | **10.5** |
| *Urad* | Ureidoimidazoline decarboxylase | | | | | 70.3 | **1.7** | 128.5 | **9.8** |
| *lncRNA* | ENSMUST00000215098.2 | 4124 | **34.4** | 3779.4 | **33.3** | 3322.7 | **4.4** | 2828.2 | **6.4** |
| *lncRNA* | E230029C05R | 221.9 | **1.9** | 203.4 | **2.9** | | | | |

Values listed in the '4d het' and '10d het' columns are mRNA read averages from RNA-seq experiments after crypt isolation and FACS sorting for GFP positive cells from $Mll1^{FC/+;\ Lgr5-eGFP-CreERT2/+}$ mice 4 days (4d) or 10 days (10d) after the final tamoxifen induction. The adjacent FD columns show Fold Down mRNA read average change from the accompanying experiments with $Mll1^{FC/FC;\ Lgr5-eGFP-CreERT2/+}$ mice.

Values listed in 'P1 het' and 'P3 het' are mRNA read averages from RNA-seq experiments with $Mll1^{FC/+;\ RC/+}$ organoids passage 1 (P1) or passage 3 (P3) after 4-hydroxy tamoxifen induction. The adjacent FD columns show Fold Down mRNA read average change from the accompanying experiments with $Mll1^{FC/FC;\ RC/+}$ organoids.

All protein coding mRNAs ≥ 1.4 FD in both 4 day or P1 het experiments were considered except those expressed less than 50 read average, which were excluded. Only genes ≥ 1.6 FD in at least two columns are shown. All data p value < 0.05 except those marked with an asterisk*. Using the same criteria, two long non-coding RNAs also qualified and are listed at the bottom.

to the RNA-sequencing mRNA reads (S5C Fig). Western blot analysis on crypts isolated from *Mll1*[FC/+; RC/+] and *Mll1*[FC/FC; RC/+] mice demonstrated downregulation of GATA4 protein (Fig 4G). Concordant with RNA-sequencing data from *Mll1*-deficient forebrain [52,53], mRNA for the chromatin co-factor, *Bahcc1* was downregulated (Table 1). BAHCC1 binds to H3K27me3 and sustains repression of target genes [54]. These data suggest that MLL1 maintains the transcriptional identity of ISCs.

H3K4me3 chromatin immunoprecipitation indicated that *Foxa1*, *Pitx1*, *Pitx2*, *Gata4*, *Onecut2*, *Zfp503/Nolz1* and *Bahcc1* promoters are direct targets for MLL1. However similar indications for *Far1*, *Hspa8* or *Ces2g* were not obtained (Fig 4H). Notably two long non-coding RNAs featured prominently in the list of downregulated transcripts (Table 1). One of these, ENSMUST00000215098.2, overlaps with the last two exons of *Jaml/Amica1* and was very strongly downregulated. However this transcript appears to play no role in the regulation of JAML expression, which remained unchanged after loss of MLL1 as evaluated by Western blot analysis (S5D Fig).

## Loss of MLL1 in ISCs provokes transcriptional changes in Paneth cells

ISCs are anchored in the crypt in close association with Paneth cells. In order to elucidate whether the transcriptional changes in ISCs influenced the neighboring Paneth cells, we analyzed Paneth cell transcriptional profiles 4 days after deletion of *Mll1* in Lgr5+ ISCs, to find 198 and 72 transcripts significantly up- or downregulated respectively (Fig 5A and S1 Data). The most significant terms associated with downregulated mRNAs relate to perturbation of protein folding and homeostasis in the endoplasmic reticulum (Fig 5B–5D). In contrast upregulated mRNAs associate with metabolic changes. Strikingly, transcripts of genes belonging to all five respiratory chain complexes were upregulated (Fig 5B–5D). Paneth cells normally utilize glycolysis with lactate as the end product whereas ISCs depend on mitochondrial oxidative phosphorylation [55]. These transcriptional changes suggest that loss of MLL1 in the stem cell compartment provokes changes in Paneth cells and indeed expression of Paneth cell marker genes was downregulated (Fig 5C and 5D).

## Skewed differentiation of organoids after loss of MLL1

To evaluate the cell-intrinsic requirement of MLL1 in the small intestine, we isolated crypts from *Mll1*[F/+; RC/+] and *Mll1*[F/F; RC/+] mice and cultured organoids [56]. After passaging, the organoids were induced with 4-hydroxy tamoxifen for 24 hours on day 2. After further passages, the *Mll1*[FC/FC; RC/+] organoids showed a gradual decrease in budding and increasingly formed round, less differentiated cyst-like spheres (Fig 6A and 6B).

To unravel the molecular basis of the intestinal organoid requirement for MLL1, transcriptome analyses of *Mll1*[FC/+; RC/+] and *Mll1*[FC/FC; RC/+] organoids were evaluated after passages 1 and 3 (P1 and P3). Using paired end 100 bp reads, 38–64 million reads per sample with high levels of uniqueness (74–81% at P1 and 68–88% at P3; S6A and S6B Fig) and comparable mapability (97%) were obtained. PCA revealed that all biological replicates clearly separated according to genotype thereby lending confidence regarding data quality (S6C and S6D Fig). To analyze DEGs, we applied DESeq2 analysis to the RNA-seq dataset. At P1 only 24 genes were up- or downregulated at a 5% FDR (Fig 6C) including an exceptional concordance with downregulated genes in the ISC RNA-seq (Table 1), namely the transcription factors *Pitx2*, *Foxa1*, *Zfp503*, *Pitx1* and *Gata4* (Table 1 and S1 Data). Other commonly downregulated genes include *Bahcc1*, *Hspa8* and *Prss12*. In contrast, no significantly upregulated genes in common between 4 day ISC and P1 organoid transcriptomes were observed, potentially reflecting the different cellular compositions and circumstances between sorted ISCs and organoids in

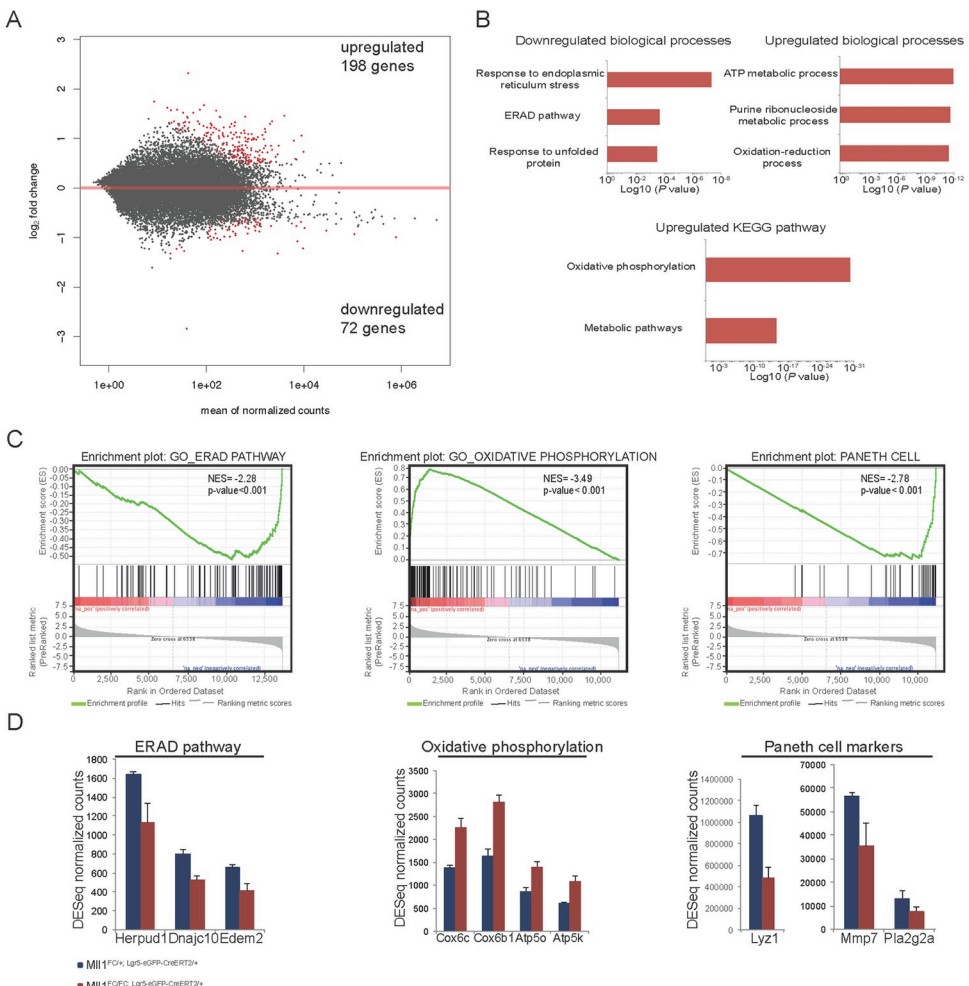

**Fig 5. RNA profiling of wt Paneth cells after deletion of *Mll1* in the neighboring ISCs.** (A) Wt Paneth cells neighboring either *Mll1^{FC/+; Lgr5-eGFP-CreERT2/+}* or *Mll1^{FC/FC; Lgr5-eGFP-CreERT2/+}* ISCs were sorted 4 days after tamoxifen induction was completed and were subjected to RNA profiling. MA plot visualizing the log2-fold change differences according to expression levels. Red dots represent significant DEGs at a 5% FDR. (B) Enriched terms of biological processes and pathways down- and upregulated using DAVID GO/BP/FAT and KEGG database. (C) GSEA shows significant negative or positive correlation of genes from the GO ERAD pathway, GO oxidative phosphorylation and Paneth cell signature gene set in wt Paneth cells neighboring either *Mll1^{FC/FC; Lgr5-eGFP-CreERT2/+}* or *Mll1^{FC/+; Lgr5-eGFP-CreERT2/+}* ISCs. The Paneth cell signature gene set originates from [87]. NES: normalized enrichment score. (D) DESeq normalized counts for selected genes differentially regulated in the ERAD pathway, oxidative phosphorylation and Paneth cell signature gene set. Mean+s.d. is shown; n = 4; p<0.05, Wald test.

culture. DAVID analysis of the P1 organoid data showed that the upregulated transcripts mainly related to diverse terms including 'response to external stimulus', 'leukocyte cell-cell adhesion' and 'cell response to chemical stimulus' (Fig 6D). In contrast, the most significant terms associated with downregulated transcripts after MLL1 removal were comparable to the downregulated terms of the sorted ISCs (Fig 4B) including 'cell fate commitment', 'negative regulation of gene expression' and 'intestinal epithelial cell differentiation' (Fig 6E). After P3, many more (1629 and 434) genes were up- and downregulated at a 5% FDR (S6E Fig). The downregulated transcripts primarily associated with metabolic changes potentially reflecting the transition to spheroids (S6F Fig). Nevertheless, most of the prominent downregulated mRNAs common to ISCs and P1 organoids were also downregulated at P3 (Table 1 and S7 Fig). The most significant terms associated with upregulated transcripts relate to 'cell surface

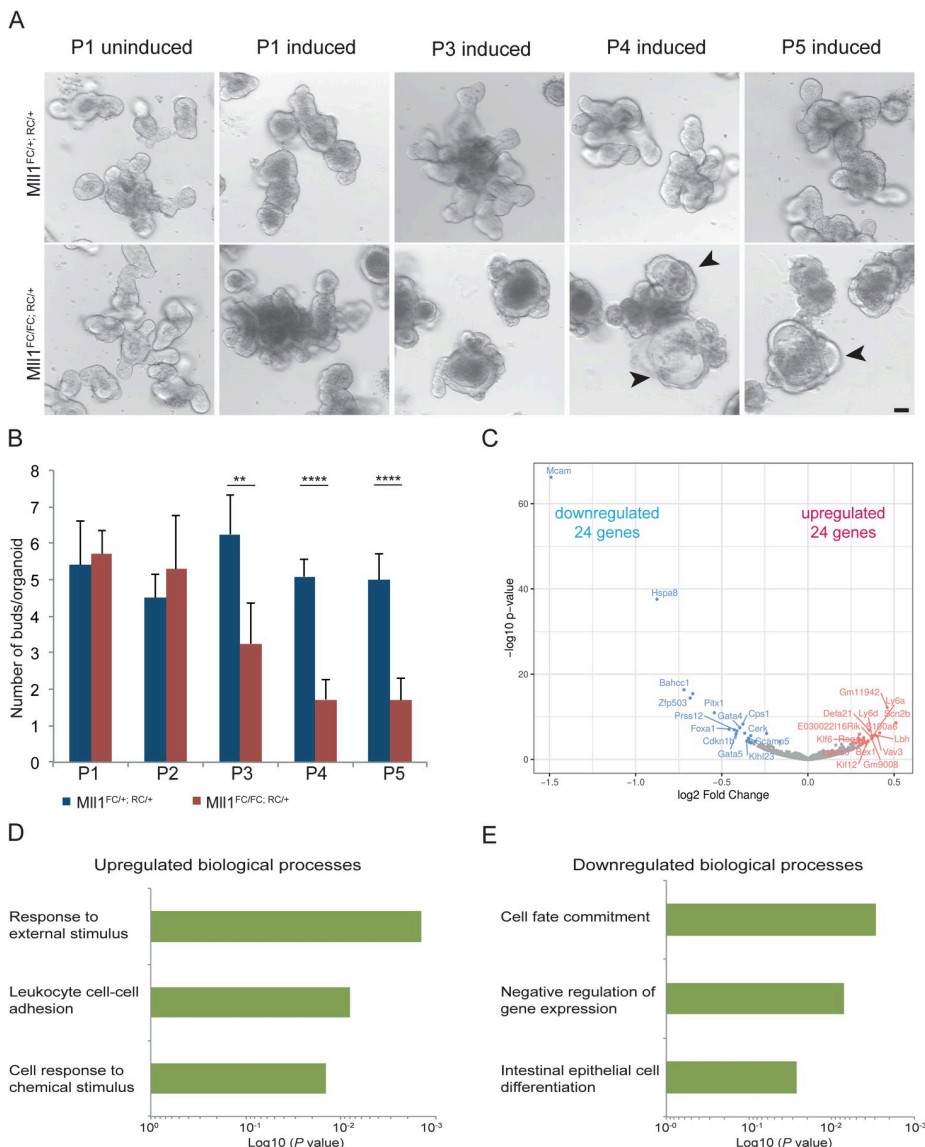

**Fig 6. *Mll1* deletion in organoids results in loss of budding and formation of spheres.** (A) Differential interference contrast (DIC) images of $Mll1^{FC/+; RC/+}$ and $Mll1^{FC/FC; RC/+}$ organoids. Organoids were induced with 4-hydroxy tamoxifen for 24 h. Upon passaging (P), the mutant starts to loose its budding morphology giving rise to an undifferentiated cyst-like appearance. Scale bar 100 μm. (B) Quantification of number of buds per organoid shown in A. Mean+s.d. is shown; *p<0.05, **p<0.01, ***p<0.001, ****p<0.0001, Student's *t* test. (C) mRNA profiling of $Mll1^{FC/+; RC/+}$ and $Mll1^{FC/FC; RC/+}$ intestinal organoids after passage 1. Volcano plot visualizing the log2-fold change differences according to expression levels. Blue and red dots represent significant down- and upregulated DEGs, respectively, at a 5% FDR. (D, E) Enriched terms of biological processes after passage 1 upregulated (D) and downregulated (E) using DAVID GO/BP/FAT.

receptor signaling pathway', 'regulation of cell communication' and 'regulation of signaling' (S6G Fig). Additionally, 'regulation of MAPK cascade' and 'Wnt signaling pathway' were also upregulated at P3 (S6G and S6H Fig). KEGG pathway analysis suggests rearrangements in the extracellular matrix of organoids with laminins (e.g. *Lama3*, *Lama5*), fibronectins (e.g. *Fn1*) and collagens (e.g. *Col4a1*, *Col4a2*) being more prevalent (S6H and S6E Fig). Notably 662 out of 1629 upregulated transcripts (41%) were only expressed in $Mll1^{FC/FC; RC/+}$ cyst like spheres indicating a change in cell type specificity (S1 Data). These observations confirm that the

requirement for MLL1 is cell intrinsic and essential for the maintenance of crypt morphology and budding in organoid cultures.

Overall the data establish that MLL1 sustains the expression of key ISC genes. Their reduced expression after loss of MLL1 leads to a loss of ISCs, consequent loss of TA cells and aberrant secretory cell differentiation biased towards goblet cells. These perturbations in intestinal homeostasis lead to wasting and death of the mouse.

## Discussion

Here we add a fourth stem cell to the known MLL1 repertoire of (i) HSCs [21,22], (ii) skeletal muscle satellite cells [27] and (iii) postnatal neural stem cells (NSCs) [28]. Additionally, MLL1 is required to sustain salivary gland and intestinal cancer stem cells mutated in the Wnt pathway [48,57,58]. Consequently, all known MLL1 functions relate to action in stem cells, which suggests that MLL1 conveys an essential stem cell property. This possibility is enhanced by comparison to the MLL1 paralogue, MLL2, whose known functions in adult mice do not relate to stem cells rather to macrophages (for lipopolysaccharide response) or fertility [59–62].

To explore the idea that MLL1 conveys a key stem cell property, we inspected the transcriptome profiles after conditional loss of MLL1 in the four adult/postnatal stem cell populations [27,28,63] (S1 Data). Some overlap between two stem cell types was observed, however no fully shared candidate regulators or gene expression programs were identified. Although deeper, more systematic, transcriptome or cell biology approaches may reveal a shared MLL1 stem cell property, the lack of concordance between MLL1 regulation of these four stem cell transcriptomes is not unexpected. Previous work with MLL1 noted that direct target genes are not shared between different cell types [63] and a similar observation was made for MLL2 [59]. That is, the regulation of gene expression by the Trithorax homologues, MLL1 and MLL2, varies depending on the cell type and is not universal.

As documented here for ISCs and intestinal organoids, once again the strongest relationship between the loss of MLL1 and cellular processes involves the downregulation of mRNAs that regulate transcription. Upon loss of MLL1, downregulation of transcription factor mRNAs include—(i) in HSCs; *Mecom*, *Prdm16*, *Pbx1*, *Eya1*, *Meis1* and *Hoxa9*; (ii) in postnatal NSCs; *Nkx2.1*, *Nkx2.3*; (iii) in satellite cells; *Pax7* and now (iv) in ISCs; *Pitx1*, *Pitx2*, *Foxa1*, *Zfp503*, *Gata4* and *Onecut2*.

How does MLL1 regulate key lineage specific transcription factors differently in different lineages? MLL1 and MLL2 are amongst the few proteins that include the CxxC zinc finger that binds unmethylated CpG dinucleotides [24,64] as well as a PHD finger that binds H3K4me3 [65]. Hence, as suggested before [59], MLL1 and 2 have the potential ability to bind CpG island promoters without the need for recruitment by sequence specific DNA binding transcription factors. This potential accords with the observation that both MLL1 and 2 appear to be bound at almost all active promoters [45,66]. Consequently additional factors are required to explain the restricted transcriptional specificities of the MLLs. Notable in this regard, PAX7 is bound to MLL1 when satellite cells are activated, and enhanced transcriptional activation from both the *Myf5* promoter, to initiate skeletal muscle replenishment, and the *Pax7* promoter itself, depends on MLL1 [27]. This suggests that key transcription factors can either acquire the ability to interact with MLL1 bound at target promoters or recruit MLL1 to target promoters, or both.

Amongst the transcription factor mRNAs identified after loss of MLL1 in ISCs, *Pitx2* is prominent. *Pitx2* was previously identified as a direct target of MLL1 in ESCs and HSCs/hematopoietic progenitor cells [67,68]. PITX2 is a homeodomain protein responsible for left-right asymmetric morphogenesis in the gut and proper positioning of the small intestine in the

body cavity [69]. Also notably identified in ISCs are *Foxa1* and *Onecut2*. Both genes, previously known as *Hnf3α* and *Hnf6α*, are expressed in all epithelia of the gastrointestinal tract from its embryonic origin into adulthood. Together with *Math1*, they are critical for goblet cell differentiation and function [70,71]. *Gata4*, which has previously been described as an MLL1 target gene [48,72], is also amongst the top downregulated mRNAs after loss of MLL1. Some aspects of the intestine specific deletion of *Gata4* in the adult mouse resemble the MLL1 phenotype described here including decreased proliferation in the crypt with increased numbers of goblet cells [73]. Also notable is downregulation of the chromatin accessory factor, BAHCC1, which has recently been shown to bind H3K27me3 and loss of *Bahcc1* expression led to loss of H3K27me3-mediated repression of target genes, thereby defining an H3K27me3 mechanism for repression that does not rely on Polycomb Repressive Complexes (PRCs) [54]. Hence maintenance of *Bahcc1* expression provides a further component for the maintenance of ISC identity by MLL1 through repression of gene expression, potentially those involved in the MAPK and Wnt signaling pathways [48].

Upon loss of MLL1, the close association between ISCs and Paneth cells in the crypt and organoids is destabilized (Fig 7). We suggest that loss of niche integrity leads to secondary alterations particularly in the Paneth cell transcriptome but also possible changes in niche properties especially to the central ISC at the bottom of the crypt, which is flanked by two Paneth cells and has long-term self-renewal potential compared to border ISCs [74]. Notably, MLL1 regulates the positional identity of NSCs [28], which may be relevant to the positional identity of the central ISC.

Deletion of *Mll1* in organoid cultures resulted in a gradual transition from branched organoids to spheroids. Under normal conditions, ISCs and Paneth cells reside in the tightly curved organoid budding regions where adherence junctions and high curvature limits stem

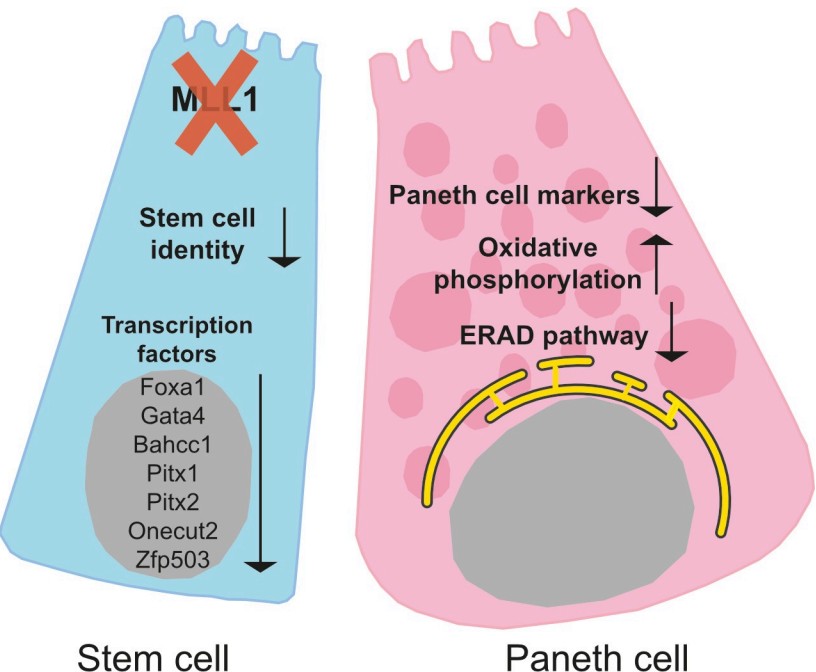

**Fig 7. MLL1 is essential for niche homeostasis.** ISCs are anchored in the crypt in close association with Paneth cells. Loss of MLL1 in ISCs causes transcriptional changes with downregulation of intestine specific transcription factors such as *Pitx1*, *Pitx2*, *Foxa1* and *Gata4* as well as chromatin accessory factor *Bahcc1*. Subsequently loss of stem cell identity perturbs Paneth cell identity accompanied with changes in metabolism and protein folding.

cell expansion [75]. Loss of MLL1 provokes cellular changes that leads to the formation of spheroids. Notably, the transition to spheroids was not accompanied by a loss of proliferation, which stands in contrast to the loss of MLL1 in the crypt. This difference may be due to the absence of mesenchymal tissue in organoids, which exhibit a complex dynamic in intestinal crypts, or possibly de-differentiation of the ISCs in the organoids to an MLL1-independent state.

Once again we report the remarkable specificity of regulated gene expression conveyed by a Trithorax homologue. As described here, loss of MLL1 resulted in loss of transcriptional regulation with subsequent significant functional consequences in the crypt. In ESCs, the expression of only one gene, *Magoh2*, entirely depends on MLL2. Removal of MLL2 resulted in the suppression of *Magoh2* expression by H3K27 methylation, followed by DNA methylation [59,76] thereby providing more evidence supporting the conclusion that a primary function of Trithorax action is to prevent Polycomb-Group repression [77]. These observations and conclusions with *Magoh2* in ESCs were recently confirmed [78]. Selective gene specific anti-repression could also explain the action of MLL2 on *Pigp* in macrophages [60] and MLL1 on *Hoxa9* in HSCs [26].

Despite a high degree of evolutionary conservation and near-ubiquitous expression, the Trithorax homologues appear to regulate only a small number of genes in cell-type specific patterns. As observed in other adult stem cells, MLL1 regulates the expression of transcription factors in ISCs that maintain stem cell identity and likely influence lineage commitment decisions. These observations lead to the attractive proposition that MLL1 is a master stem cell regulator. However, a unifying mechanism for this proposition remains to be identified.

## Methods and materials

### Ethics statement

Experiments were performed in accordance with German animal welfare legislation, and were approved by the relevant authorities, the Landesdirektion Dresden.

### Targeting constructs

The targeting construct for *Mll1* was generated using recombineering (S1A Fig) employing an engrailed-intron-splice-acceptor-IRES-LacZ-Neomycin-polyA cassette flanked by FRT sites [44]. The critical exon 2, which upon deletion results in a frameshift and a premature stop codon in exon 3, was flanked by loxP sites.

### Gene targeting and generation of conditional knockout mice

Gene targeting in R1 embryonic stem cells (ESCs) was performed as described [79]. Correct integration in the *Mll1* locus was confirmed by Southern blot analysis using 5' and 3' external probes and a LacZ internal probe (S1B and S1C Fig). Two correctly targeted ESC clones were injected into blastocysts and gave rise to several chimeras, which were able to establish germ line transmission. *Mll1$^{A/+}$* mice were crossed to the *hACTB-Flpe* line to generate *Mll1$^{F/+}$* mice. *Mll1$^{A/+}$* and *Mll1$^{F/+}$* mice were backcrossed at least six generations to *C57BL/6JOlaHsd* mice. Subsequently, those mice were crossed to the *Rosa26-Cre-ERT2* (*RC*) line [42] to generate conditional, tamoxifen-inducible *Mll1$^{F/+; RC/+}$* mice. *Mll1$^{F/+}$* mice were bred with *Lgr5-eGFP--CreERT2* and *Villin-CreER$^{T2}$* (*Vil-Cre-ERT2*) [30,49] mice. Primers for genotyping are provided in S2 Table.

## Tamoxifen

Tamoxifen (Sigma Aldrich, T5648) was given to at least 10-week old mice by gavage (4.5 mg per day) for six days with three days break in between [80]. For RNA-Sequencing experiments, $Mll1^{F/+; Lgr5-eGFP-CreERT2/+}$ and $Mll1^{F/F; Lgr5-eGFP-CreERT2/+}$ mice received 1 mg tamoxifen via intraperitoneal (IP) injection for 3 consecutive days. To obtain $Mll1$-deficient crypts $Mll1^{F/F; RC/+}$ and $Mll1^{F/+; RC/+}$ mice received tamoxifen (4.5 mg per day) via gavage for four days and were sacrificed seven days after the final gavage. Intestinal organoids were induced on day 2 after splitting using 800 nM 4-hydroxy tamoxifen (Sigma Aldrich, H7904) for 24 h.

## Bone marrow transplantation

$Mll1^{F/+; RC/+}$ and $Mll1^{F/F; RC/+}$ recipient (CD45.2) mice were lethally irradiated with 8.5 Gy (X-ray source MaxiShot from Yxlon). Bone marrow cells from B6.SJL mice (CD45.1) were prepared by crushing femurs and tibiae with a mortar and pestle in ice-cold PBS supplemented with 5% fetal bovine serum. Red blood cells were removed with ACK lysis buffer (Thermo Fisher Scientific). $1 \times 10^6$ lineage depleted (Lin$^-$) bone marrow cells were injected into the retro-orbital venous plexus. Animals were maintained on water containing 1.17 mg/ml neomycin (Merck) for three weeks after irradiation. Complete donor cell engraftment of wt CD45.1$^+$ cells was confirmed by flow cytometry on peripheral blood with antibodies directed against the following murine antigens (clones given in brackets): CD45.1 (A20), CD45.2 (104), CD11b (M1/70), Gr-1 (RB6-8C5). Stably engrafted mice were fed six times with tamoxifen 30 weeks after transplantation. FACS analysis for KSL-Slam enriched HSCs was done with antibodies directed against the following murine antigens: CD3 (2C11; 17A2), CD11b (M1/70), CD16/32 (93), CD19 (eBio 1D3), CD34 (RAM34), CD45.1 (A20), CD45.2 (104), CD45R (RA3-6B2), CD117 (2B8), CD135 (A2F10), Gr-1 (RB6-8C5), Nk1.1 (PK136), Ter119 (Ter119, all eBioscience), CD11b (M1/70) and CD45.1 (A20, all BD Pharmingen), Sca-1 (D7), CD48 (HM48-1) and CD150 (TC15-12F1, all BioLegends). Lin$^-$ cells were identified by lack of CD3, CD11b, CD19, CD45R, Gr-1, Nk1.1 and Ter119 expression.

## 5-Bromo-2-deoxyuridine (BrdU) assay

Mice were injected IP with BrdU (0.6 mg/10 g body weight in sterile PBS) and sacrificed 2 h later. The jejunum was dissected in cold PBS and processed for immunohistochemistry.

## Histochemistry and immunohistochemistry

Mouse intestine was flushed gently with cold PBS. Embryos were dissected from plugged mice on the respective gestational stage and placed in PBS. Intestine and embryos were fixed in 4% paraformaldehyde overnight. Dehydration and paraffin infiltration utilized the Paraffin-Infiltration-Processor (STP 420, Zeiss). Dehydrated tissues were embedded in paraffin (Paraffin Embedding Center EG1160, Leica) and 5 μm sections were prepared. Sections were deparaffinized in xylene, rehydrated through a series of alcohols, stained, dehydrated, and mounted. Femur sections were stained with Giemsa according to standard protocols. For the basic evaluation of the intestine hematoxylin and eosin (H&E) stain was performed. Periodic acid-Schiff (PAS) stain and alcian blue were used to identify goblet cells. For enterocytes, alkaline phosphatase stain was performed using two different methods. Sections were either incubated with Red Alkaline Phosphatase Substrate (Vector red, Vector Laboratories) for 10 minutes (min) or with nitroblue tetrazolium/5-bromo-4-chloro-3-indolyl phosphate solution for 30 min at room temperature (RT). For immunohistochemistry, antigen retrieval was performed by microwaving slides in 10 mM citrate buffer (pH 6.0) for 12 min (Microwave RHS 30, Diapath).

Endogenous peroxidases were quenched with 0.3% $H_2O_2$ in methanol. Sections were incubated in blocking serum (5% goat serum) for 1 hr at RT followed by overnight incubation with primary antibodies (S3 Table) at 4˚C. Following incubation with the secondary antibody (S3 Table) the immune peroxidase was detected using a Vectastain ELITE ABC kit (Vector) and visualized with a solution of diaminobenzidine (Sigma Aldrich) in the presence of 0.01% $H_2O_2$. All sections were counterstained with hematoxylin or alcian blue. Images were collected with an Olympus WF upright microscope and analyzed using the MetaMorph Microscopy Automation and Image Analysis Software.

### *In situ* hybridization

The protocol for *in situ* hybridization was modified from [81]. Briefly, 8 μm thick sections were rehydrated. The sections were treated with 0.1 N HCl and proteinase K. Slides were postfixed and sections were then demethylated with acetic anhydride and prehybridized. Hybridization was done with 2 μg/ml digoxigenin (DIG)-labeled *Olfm4* RNA probe for 24 h at 65˚C. Slides were washed and incubated with blocking solution for 1 hr. The sections were incubated with anti-DIG-alkaline phosphatase conjugate overnight at 4˚C. Slides were washed and developed with BM purple.

### Crypt isolation

The jejunum was harvested from mice, flushed with ice cold PBS to remove any faecal content and cut open longitudinally. The tissue was placed lumen side up on a petri dish and the villi were removed by gently scraping the tissue using a glass cover slip. The tissue was cut into 2–4 cm pieces and was washed several times with ice-cold PBS to remove residual villi fragments. Tissues were transferred into a fresh tube containing 15 ml of 2 mM EDTA/PBS chelation buffer and placed on a rotating wheel for 30 min at 4˚C. The crypts were then detached from the basal membrane by vigorous shaking in 5% FCS/PBS solution. The suspension was filtered with a 100 μm cell strainer followed by a 70 μm cell strainer. Isolated crypts were centrifuged at 800 rpm for 5 min at 4˚C. The final fraction consisted of pure crypts and was used for organoid culture or single cell dissociation.

### Organoid culture

Purified crypts were resuspended in 10 ml DMEM/F12 (Life Technologies). 10 μl of the crypt suspension was used to count the number of crypts under the microscope. The pelleted crypts were resuspended in Matrigel matrix (Corning) at desired crypt density. Approximately, 400 crypts in 25 μl Matrigel matrix were seeded per well in a pre-warmed 24-well plate and incubated for 15 min at 37˚C until the Matrigel matrix solidified. Then, 400 μl of IntestiCult Organoid Growth Medium (STEMCELL Technologies) was added to each well. Organoids were cultured at 37˚C in a 5% $CO_2$ incubator and maintained in culture for 5 days before being passaged and split for experimental procedures. The growth medium was replaced every 2–3 days. Organoids at passage 1 and 3 on day 5 were washed with PBS and total RNA was extracted using Trizol.

### Flow cytometry

ISCs were initially characterized and identified with the use of an *Lgr5-eGFP-CreERT2* knockin allele [30]. $Mll1^{F/+; Lgr5-eGFP-CreERT2/+}$ and $Mll1^{F/F; Lgr5-eGFP-CreERT2/+}$ littermates (n = 4) were given tamoxifen via IP injections for 3 consecutive days and were dissected 4 and 10 days later. Crypts were dissociated into single cells with TrypLE Express (Thermo Fisher Scientific) for 30

min at 37˚C. Dissociated cells were passed through 70 μm cell strainer and washed with 5% FCS/PBS. Cells were stained with antibodies with the following antibodies for 45 min on ice: anti-mouse CD24 PE (clone M1/69), anti-mouse CD326 (EpCAM) APC (clone G8.8), anti-mouse CD45 Alexa-Fluor 700 (clone 104). Sorting was performed on a FACS Aria III cell sorter (BD). After scatter discrimination to remove doublets the cell suspension was negatively selected with SYTOX blue dead cell stain and anti-CD45 to remove dead and hematopoietic cells, respectively. The cells where then positively selected with anti-EpCAM to enrich for intestinal epithelial cells. According to [82] CD45$^-$, EpCAM$^{high}$, CD24$^{med}$ and GFP$^{high}$ characterized ISCs and CD45$^-$, EpCAM$^{high}$, CD24$^{high}$ and GFP$^-$ characterized Paneth cells (S3A and S3B Fig). *Mll1* recombination via PCR on the sorted populations confirmed deletion of *Mll1* solely in the stem cell compartment (S3C Fig).

## RNA sequencing

300 intestinal stem and Paneth cells were sorted into 2 μl of nuclease free water with 0.2% Triton-X 100 and 4 U murine RNase Inhibitor (NEB). RNA was reverse transcribed (Invitrogen) and cDNA amplified using Kapa HiFi HotStart Readymix (Roche). The cDNA quality and concentration was determined with the Fragment Analyzer (Agilent). Samples were subjected to library preparation (TruePrep DNA library Prep Kit V2 for Illumina, Vazyme). Libraries were purified followed by Illumina sequencing on a Nextseq500 with a sample sequencing depth of 30 million reads on average.

mRNA was enriched from 500 ng total RNA from organoids of the respective genotype by poly-dT enrichment using the NEBNext Poly(A) mRNA Magnetic Isolation Module according to the manufacturer's instructions. The polyadenlyated RNA fraction was eluted in first strand cDNA synthesis buffer (NEBnext, NEB). After chemical fragmentation samples were directly subjected to the workflow for strand specific RNA-seq library preparation (Ultra II Directional RNA Library Prep, NEB). Libraries were purified, quantified using the Fragment Analyzer (Agilent) and sequenced on an Illumina Novaseq 6000 with 2x 100 bp reads using a S4 flowcell to an average depth of 50 million read pairs.

The short reads were aligned to the mm10 transcriptome with GSNAP (2018-07-04 and 2020-12-16) and a table of read counts per gene was created based on the overlap of the uniquely mapped reads with the Ensembl Gene annotation (version 92 and 98), using feature-Counts (version 1.6.3 and 2.0.1). Normalization of the raw read counts based on the library size and testing for differential gene expression between the different genotypes was performed using the DESeq2 R package (version 1.24.0 and 1.20.0). Genes with an adjusted p-value (padj)$\leq$ 0.05 were considered as significantly differentially expressed accepting a 5% FDR. To identify enrichment for particular biological processes and pathways associated with the DEGs, the DAVID GO/BP/FAT and KEGG database [83] was used. Gene set enrichment analysis was performed using GSEA software from the Broad Institute [84].

## Reverse transcription and quantitative PCR (qRT-PCR) analysis

RNA from sorted cells was extracted using Trizol (Sigma-Aldrich) and reverse transcribed using AffinityScript Multiple Temperature cDNA Synthesis Kit (Agilent Technologies). Real-time quantitative PCR was performed with GoTaq qPCR Master Mix (Promega) by Mx3000P QPCR System (Agilent Technologies). Ct values were normalized against *Rpl19*. Primer sequences and length of the amplified products are given in S2 Table. Fold differences in expression levels were calculated according to the $2^{-\Delta Ct}$ method [85].

## ChIP-qPCR

ChIP-qPCR was performed as previously described [4,45,86]. In brief, equal amounts of crypts were crosslinked by adding formaldehyde followed by addition of glycine to quench the formaldehyde. After a wash with ice cold PBS the cells were harvested and lysed. Sonication was performed with a water bath sonicator (BioRuptor, Diagenode) for 20 min at high power, 30 seconds ON, 30 seconds OFF resulting in DNA fragments of 200–600 bp. Immunoprecipitation was performed using an antibody against H3K4me3 (ab8580; Abcam). A mock control without addition of antibody was always performed in parallel. After washing the immune complexes were eluted with 500 μl 1% SDS, 0.1 M NaHCO$_3$ for 30 min, NaCl was added to a final concentration of 200 mM and samples were de-crosslinked overnight at 65˚C. Samples were extracted with phenol/chloroform/isoamylalcohol, isopropanol-precipitated in the presence of 20 μg glycogen and resuspended in water. The immunoprecipitated DNA was amplified by qPCR using primers flanking the promoter region (S2 Table). ChIP-qPCR data were normalized with percentage of input analysis. This method represents the amount of DNA pulled down by using the antibody of interest in the ChIP reaction, relative to the amount of starting material (input sample).

## Western blot analysis

Crypt and organoid pellets were resuspended in ice-cold RIPA buffer supplemented with 1x cOmplete Protease Inhibitor Cocktail (Roche) and PMSF for 15 min on ice, centrifuged (13200 rpm, 4˚C, 15 min) and supernatant was collected. Whole cell extracts were separated by SDS-PAGE (10% Tris-glycine gel) and transferred to a PVDF membrane. Blots were probed with primary antibodies (S3 Table).

## Quantification and statistical analysis

Data is presented as mean and error bars indicate standard deviation (s.d.) unless otherwise indicated. Statistical details of the experiments can be found in the figure legends. Graphs and statistics were generated with GraphPad Prism software (v6.0) and Microsoft Excel. Significance (p-values) for Kaplan-Meier graphs was determined by Mantel-Cox test. Significance for differential gene expression (p-values < 0.05) was determined with Wald test. Significance for the breeding statistics was determined with Chi-square test. All other graphs are computed with two-tailed Student's *t* test. N indicates the numbers of independent biological replicates per experiment unless otherwise indicated.

## Supporting information

**S1 Fig. *Mll1* gene targeting, embryonic phenotype and aspects of expression. (A)** Diagram of the *Mll1* gene with numbered exons and the multipurpose allele (*Mll1$^A$*). This allele is converted to *Mll1$^F$* upon FLP recombination. Cre recombination leads to excision of the frame-shifting exon 2 generating the conditional mutant allele (*Mll1$^{FC}$*). Genotyping primers are depicted for the downstream loxP site (loxP1 –loxP2) and for Flp recombination (Flp se–loxP2). SA = splice acceptor, IRES = internal ribosome entry site, pA = polyadenylation signal, lacZ-neo = β-galactosidase and neomycin resistance gene, * depicts premature stop codon. **(B)** Schematic representation of the Southern blot strategy. For identifying correct targeted events in the *Mll1* locus, Southern blot analysis employed 5' (blue box), 3' (red box) and internal LacZ (green box) probes. **(C)** Southern blot analysis with 5' external probe (wt and correctly targeted ESC clone). Southern blot analysis with 3' external and LacZ internal probes (wt ESCs and organs from an adult *Mll1$^{A/+}$* mouse). **(D)** Dissected embryos from *Mll1$^{A/+}$* intercrosses at

E12.5. *Mll1*^A/A^ embryos had a pale liver (marked by arrow). **(E)** Antibody staining (brown) shows that MLL1 is expressed in crypts and TA compartment of the small intestine but is absent in the villus (hematoxylin, purple). Scale bar 50 μm. **(F)** Normalized RNA-sequence counts for *Mll1/Kmt2a*, *Mll2/Kmt2b*, *Mll3/Kmt2c*, *Mll4/Kmt2d*, *Setd1a/Kmt2f* and *Setd1b/Kmt2g* in ISCs (eGFP^high^) and Paneth cells sorted from *Lgr5-eGFP-CreERT2* mice. Mean+s.d. is shown; n = 4; *p<0.05, **p<0.01, ***p<0.001, ****p<0.0001, Student's *t* test. **(G)** Antibody stainings of H3K4me1, H3K4me2 and H3K4me3 are comparable in *Mll1*^FC/+; RC/+^ and *Mll1*^FC/FC; RC/+^ intestinal sections. Scale bars are 100 μm.
(TIF)

**S2 Fig. Without bone marrow transplantation, *Mll1* deletion recapitulates the BMTx *Mll1* mutant phenotype. (A)** Kaplan-Meier analysis for the onset of diarrhea. Tamoxifen was given by gavage for 6 days to *Mll1*^F/+; RC/+^ (n = 11) and *Mll1*^F/F; RC/+^ (n = 11) mice. The first day of tamoxifen gavage was day zero. While all mice with the genotype *Mll1*^FC/+; RC/+^ remained healthy, all *Mll1*^FC/FC; RC/+^ mice developed diarrhea with a median of 11 days. **(B)** Antibody stain (left panels) and *in situ* hybridization (right panels) to visualize OLFM4/*Olfm4* in intestinal sections. Arrowheads point towards ISCs. Scale bar 100 μm. **(C)** Proliferative activity visualized by both Ki67 stain and BrdU incorporation in intestinal sections. Arrowheads point towards proliferative ISCs. Scale bars are 50 μm. **(D)** PAS staining and GOB5 antibody stain to visualize goblet cells in intestinal sections. Scale bars are 100 μm. **(E)** Chromogranin A and alkaline phosphatase staining to visualize enteroendocrine cells and enterocytes respectively. Arrows point to enteroendocrine cells (brown cytoplasmic stain) in the villi. Blue enterocytes covering the villi are marked by arrowheads. Scale bars are 100 μm for chromogranin A and 50 μm for alkaline phosphatase. **(F)** Nuclear β-catenin is comparable between the two different genotypes. Arrowheads point at β-catenin positive nuclei. Scale bar is 50 μm. **(G)** Left and middle panels; antibody stainings to visualize OLFM4 and Ki67 expression in *Mll1*^FC/FC; Vil-Cre-ERT2/+^ intestine compared to controls, four weeks after the first gavage. Hematoxylin was used as a counterstain for OLFM4 and alcian blue was used for Ki67 IHC. Right panel; alcian blue staining of goblet cells with NFR to stain nuclei, four weeks after the first gavage. Scale bars are 100 μm.
(TIF)

**S3 Fig. FACS gating strategy to sort ISCs and Paneth cells.** Flow sorting on **(A)** *Mll1*^FC/+; Lgr5-eGFP-CreERT2/+^ and **(B)** *Mll1*^FC/FC; Lgr5-eGFP-CreERT2/+^ single cell suspension of crypts. Briefly, the consecutive gating steps were applied: (i)–(iii) Definition of the population of interest by exclusion of debris based on size (FSC), granularity (SSC) and the selection for single cells; (iv) Exclusion of dead cells that incorporated the nucleic acid stain SYTOX blue; (v) Depletion of CD45^pos^ population; (vi) Definition of Paneth (EpCAM^high^/CD24^high^) cell population by plotting EpCAM vs CD24 fluorescence; (vii) EpCAM^high^/CD24^med^ cell population was gated to discriminate the stem cell population (GFP^high^). **(C)** Stem cells (SC) and Paneth cells (PC) from *Mll1*^FC/+; Lgr5-eGFP-CreERT2/+^ and *Mll1*^FC/FC; Lgr5-eGFP-CreERT2/+^ mice 4 days after tamoxifen induction were checked for recombination. Left panel; PCR genotyping was using primers (as shown in S1A Fig) upstream of the 5' FRT site and downstream of the 3' loxP site identified the *Mll1*^F^ band at 1084 bp, the wild type band a 933 bp and the *Mll1*^FC^ band at 186 bp. Right panel; primers flanking the 3' loxP site identified the *Mll1*^F^ band at 297 bp and the wild type band at 251 bp.
(TIF)

**S4 Fig. Alignment and quality of the sequenced data. (A)** ISCs and Paneth cells were analyzed from control (*Mll1*^FC/+; Lgr5-eGFP-CreERT2/+^) (ctrl) (n = 4) and *Mll1*^FC/FC; Lgr5-eGFP-CreERT2/+^

(n = 4) (KO) mice. Mapability of reads for sorted ISCs 4 days after tamoxifen induction was completed. **(B)** Mapability of reads for sorted Paneth cells 4 days after tamoxifen induction was completed. **(C)** Mapability of reads for sorted ISCs 10 days after tamoxifen induction was completed. ISCs were analyzed from control ($Mll1^{FC/+; Lgr5-eGFP-CreERT2/+}$) (ctrl) (n = 4) and $Mll1^{FC/FC; Lgr5-eGFP-CreERT2/+}$ (n = 3) (KO) mice. **(D)** Principal-component analysis (PCA) was performed on Paneth cell and ISC samples sorted 4 days after tamoxifen. PCA is based on mRNA changes for the top 500 most diverse genes of stem cell (SC) and Paneth cell (PC) samples in comparison to published datasets for Lgr5+ SC [51] and CD24+ PC [37].
(TIF)

**S5 Fig. ISCs lacking MLL1 loose their cellular identity. (A) (B)** GSEA shows significant negative or positive correlation of genes from the stem (A) and goblet cell (B) signature gene set in $Mll1^{FC/FC; Lgr5-eGFP-CreERT2/+}$ ISCs compared to control ISCs 10 days after tamoxifen induction was completed. The signature gene sets originate from [87]. NES: normalized enrichment score. **(C)** To validate RNA-seq results qRT-PCR was performed for selected genes on cDNA from $Mll1^{FC/+; Lgr5-eGFP-CreERT2/+}$ and $Mll1^{FC/FC; Lgr5-eGFP-CreERT2/+}$ sorted stem cells 4 days after tamoxifen induction was completed. Mean+s.d. is shown; n = 3; *p<0.05, **p<0.01, Student's *t* test. **(D)** Western blot analysis shows no change in JAML protein levels in $Mll1^{FC/+; RC/+}$, $Mll1^{FC/FC; RC/+}$ and $Mll1^{F/F; RC/+}$ organoids (n = 2). GAPDH is the loading control.
(TIF)

**S6 Fig. mRNA profiling of $Mll1^{FC/+; RC/+}$ and $Mll1^{FC/FC; RC/+}$ organoids. (A, B)** Intestinal organoids were analyzed from $Mll1^{FC/+; RC/+}$ (ctrl) (n = 3) and $Mll1^{FC/FC; RC/+}$ (KO) (n = 3) mice. Mapability of reads after passage 1 (A) and passage 3 (B). **(C, D)** Principal-component analysis (PCA) was performed on intestinal organoids after passage 1 (C) and passage 3 (D). PCA is based on mRNA changes for the top 500 most diverse genes. **(E)** mRNA profiling of $Mll1^{FC/+; RC/+}$ and $Mll1^{FC/FC; RC/+}$ intestinal organoids after passage 3. Volcano plot visualizing the log2-fold change differences according to expression levels. Blue and red dots represent significant down- and upregulated DEGs, respectively, at a 5% FDR. **(F-H)** Enriched terms of biological processes and pathways downregulated (F) and upregulated (G, H) using DAVID GO/BP/FAT and KEGG database.
(TIF)

**S7 Fig. Differential gene expression in $Mll1^{FC/+; RC/+}$ and $Mll1^{FC/FC; RC/+}$ organoids.** RNA-seq tracks showing the read coverage of different gene loci derived from $Mll1^{FC/+; RC/+}$ (top three rows) and $Mll1^{FC/FC; RC/+}$ (bottom three rows) intestinal organoids after passage 3. Gene diagrams are depicted below. The tracks were generated with the Integrative Genomics Viewer.
(TIF)

**S1 Table. Embryonic lethality of $Mll1^{A/A}$ embryos.**
(PDF)

**S2 Table. Primers for genotyping, qRT-PCR and ChIP-qPCR.**
(PDF)

**S3 Table. Antibodies for immunohistochemistry staining and Western blot.**
(PDF)

**S1 Data. List of up- and downregulated transcripts from sorted stem cells, sorted Paneth cells and organoids with a p value<0.05.**
(XLSX)

## Acknowledgments

We thank Mandy Obst, Isabell Kolbe, Heike Petzold and Stefanie Weidlich for excellent technical assistance. We also thank the Biomedical Services (BMS) of the Max Planck Institute of Molecular Cell Biology and Genetics, Dresden for the excellent service and technical assistance. We are grateful to Prof. Sebastian Zeissig (CRTD, Dresden) for helpful advice. We thank Katja Schneider and the FACS facility of the Center for Molecular and Cellular Bioengineering (CMCB) for providing assistance with flow cytometry. We thank Mathias Lesche for help with bioinformatic analysis. The Advanced Imaging Facility, a core facility of the CMCB Technology Platform at TU Dresden assisted this research.

## Author Contributions

**Conceptualization:** Neha Goveas, Claudia Waskow, Julian Heuberger, Konstantinos Anastassiadis, A. Francis Stewart, Andrea Kranz.

**Data curation:** Neha Goveas, Dimitra Alexopoulou, Andreas Dahl, A. Francis Stewart, Andrea Kranz.

**Formal analysis:** Neha Goveas, Claudia Waskow, Qinyu Zhang, Dimitra Alexopoulou, Andreas Dahl, A. Francis Stewart, Andrea Kranz.

**Funding acquisition:** A. Francis Stewart, Andrea Kranz.

**Investigation:** Neha Goveas, Claudia Waskow, Kathrin Arndt, Julian Heuberger, Walter Birchmeier, Konstantinos Anastassiadis, Andrea Kranz.

**Methodology:** Neha Goveas, Qinyu Zhang.

**Resources:** Julian Heuberger.

**Supervision:** Claudia Waskow, A. Francis Stewart, Andrea Kranz.

**Validation:** Neha Goveas, Andrea Kranz.

**Visualization:** Neha Goveas.

**Writing – original draft:** Neha Goveas, A. Francis Stewart, Andrea Kranz.

**Writing – review & editing:** Neha Goveas, A. Francis Stewart, Andrea Kranz.

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
