## [Decision Letter · Decision Letter 0]

29 Dec 2020

Dear Dr Kranz,

Thank you very much for submitting your Research Article entitled 'MLL1 is required for maintenance of intestinal stem cells and the expression of the cell adhesion molecule JAML' to PLOS Genetics.

The manuscript was fully evaluated at the editorial level and by independent peer reviewers. All three reviewers see merit in the study and find the results interesting. However, they also raise several important concerns which need to be addressed. In some cases, clarification of experimental approaches is needed. In other cases, a deeper discussion of key points raised by the reviewers would be helpful. In addition, the reviewers also raise several points that will likely require additional work and/or analysis. Several points raised that are of particular importance include: Mll1 binding and H3K4 methylation profiles at key gene targets in relevant cell types; better characterization of the VillinCreER and Lgr5CrerER models; and some effort towards better understanding Mll1 targeting and the relative role of Mll1 mediated H3K4 methylation at gene targets in relevant cell types. Based on the reviews, we will not be able to accept this version of the manuscript, but we would be willing to review a much-revised version. We cannot, of course, promise publication at that time.

If you decide to revise the manuscript for further consideration at PLOS Genetics, please aim to resubmit within the next 60 days, unless it will take extra time to address the concerns of the reviewers, in which case we would appreciate an expected resubmission date by email to plosgenetics@plos.org.

[LINK]

We are sorry that we cannot be more positive about your manuscript at this stage. Please do not hesitate to contact us if you have any concerns or questions.

Yours sincerely,

Tom Milne

Guest Editor

PLOS Genetics

Wendy Bickmore

Section Editor: Epigenetics

PLOS Genetics

Reviewer's Responses to Questions

**Comments to the Authors:**

Reviewer #1: The submission by Goveas builds on the interest of the group on the role of Mll1 (actually Kmt2a) an important methyltransferase that methylates Histone 3, lysine 4 residues (H3K4) to epigenetically regulate gene expression. Here the authors investigate the role of Mll1 in maintaining intestinal stem cells. Their interest arises from other tissue stem cells systems in which Mll1 has a central role including, most notably, haematopoietic but also neural and muscle satellite stem cells. In establishing Mll1 as a master regulator of stem cells the authors wish, at least in part, to perform a comparative analysis across different tissues and identify common themes or mechanism of action.

The work described does persuade that Mll1 has an important role in maintaining intestinal stem cells. However the authors account is overly simplified. They approach the topic in a blinkered way that provides an incomplete account of the work described to provide a narrative that is phenomenological with little insight into mechanism.

1. The gene name is now Kmt2a, should this not be adopted throughout?

2. The mouse used is described as though for the first time. Yet different descriptions of the floxed allele are confusing:

Line 160 refers to a likely null allele while line 167 refers to the same floxed allele and both claims are supported by ref 6 of the manuscript. Accepting the latter:why is much of the characterisation of the parental ‘A’ allele line needed if the floxed line derived from it by Flp recombination is validated?

The nomenclature used to describe the different recombined state of the modified allele is confusing. A floxed allele would often be presented as the gene name with ‘f/f’ or ‘f/wt’ in superscript. Then recombination would be designated by ‘delta’or ‘rec’. Instead here the authors use F and C to designate that the allele has already undergone Flp and/or Cre recombination.

The Flp recombination being constantly referred to implies its use in the experiments described but in fact this done once from the parental ‘A’ allele to generate what is essentially floxed conditional Mlll1 KO.

3. The early part of the manuscript documents an intestinal phenotype for Mll1 deletion by first reconstituting the bone marrow after lethal dose irradiation with wild type cells and then mediating Mll1 deletion body wide with tamoxifen. This is an inelegant solution, the gut is probably the next most sensitive tissue to irradiation so the phenotype is being described in an already damaged tissue system. The mice go down within 14 days or so with diarrhoea. The intestinal characterisation lacks key details of time post induction that analyses were done, whether the mice analysed were symptomatic or not and lacks definitive evidence that failure of the epithelial lining is the principle cause of the phenotype (as opposed, say, to compromised immune monitoring due the BM reconstitution and irradiation).

Why is colon not included in this or subsequent analyses? Was there a phenotype?

4. The subsequent validation with the more straightforward intestinal deletion of Mll1 using VillinCreER mice only partially reassures because the analysis is treated as confirmatory and is superficial. What is the outcome of Mll1 deletion; can mice be maintained long term? Why is a survival analysis not performed? When was weight loss observed? When post-induction were the analyses shown performed?

Overall there does seem to be a phenotypic downregulation of stem cell and proliferative markers and a skew in lineage composition towards goblet cells following Mll1 deletion. Assigning significance to these observations requires a better understanding of the temporal sequence leading to these changes, how they impact on stem cell function (loss of marker doesn’t necessarily equate with loss of function) and mouse survival.

5. The authors move to the Lgr5 creER model as this allows stem cells to be directly isolated due to the GFP allele in the Lgr5 locus. There seems something of a missed opportunity here as the Lgr5CrerER mice are well known to be mosaic in their expression. The longer term phenotypically consequences of Mll1 loss could have been determined with mice remaining viable due to incomplete loss of Mll1.

Also the assumption in looking at 4 and 10 day post the last dose of tamoxifen (day 7 and 13 post first injection) is that there is no phenotypic consequence yet apparent follow Mll1 ablation. This is important, is Lgr5GFP downregulated at these times? How might that impact on the analysis that follows? Perhaps the gating strategy is not randomly sampling the stem cell pool but collecting a subpopulation of Lgr5GFP expressing cells. An in situ analysis is required here.

The mechanistic insights come entirely from transcriptional changes following Mll1 ablation in Lgr5 stem cells. JAML is a potentially interesting mediator and a number of transcription factors are identified. As the authors point out deletion of Jaml in the intestine replicates much of the phenotype they describe for Mll1. However, some reassurance is needed that these candidate mediators, especially the transcription factors not known as Mll1 target genes are not altered as a secondary consequence of a developing phenotype (eg fewer enterocytes, elevated goblet cells).

6. The organoid experiments might have offered a good functional test of impaired self renewal capacity of stem cells following Mll1 ablation. However, the phenotype here did not reflect that in vivo. The cystic structures formed seem reminiscent of the spheroids that form when levels of wnt signalling are elevated.

The authors could try seeding Mll1 deficient cells or crypts and see if they have impaired seeding efficiencies.

As it stands it is hard to know how to interpret these experiments

Reviewer #2: The manuscript by Goveas used a temporally and/spatially controlled Mll1 knockout mouse model to show that Mll1 is required for maintenance of intestinal stem cells (ISCs). They first show that global deletion of Mll1 in adult mouse with wild-type hematopoietic system (by transplanted bone marrow) suffer from diarrhea and wasting, indicating functional defects in the intestinal system. Morphological, immunohistochemistry, and RNA-seq together suggest reduced ISC identity and cell proliferation, as well as increased goblet cells. Similar conclusions are confirmed by intestinal epithelium-specific KO and also using cultured organoids. They also identified a limited number of genes that altered expression levels upon Mll1 loss in ISCs. Interestingly, in addition to a number of transcription factors, a rather specific gene, JAML, was found to be particularly dependent on Mll1 in ISCs. This is similar to some of their previous studies in which they found Mll1 or Mll2 is particularly important for a few specific target genes. They also propose that Mll1 is likely a master stem cell regulator, considering its importance of a number of different somatic stem cells.

This work is original, and the methodologies are rigorous. The data are substantial for supporting the conclusions. This work expands our understanding of Mll1 as a key regulator of adult stem cells.

It is important to do co-staining of Ki67 with certain cell identity markers to show what types of cells show reduced proliferation.

It remains a mystery how Mll1/2 is specifically required for just a few target genes in a tissue-specific manner. It will be helpful to determine Mll1 binding and H3K4 methylation at the affected genes using relevant cells from animals or organoids. It could be argued that previous reports show Mll1 binds to nearly all active gene promoters and the H3K4 methylation activity of Mll1 does not seem to be the critical for certain physiological processes. However, this is a new system, and unaffected H3K4 methylation at the affected genes will help rule out Mll1’s methylation activity as a key mediator in maintaining these genes. If cell number is a limiting factor for ChIP, the authors may want to consider the newly developed CUT&RUN, which seems easier, requires far less number of cells, and have a few other advantages compared to ChIP. Any effort toward elucidating the direct effect will be appreciated.

Reviewer #3: Methylation of histone H3 on lysine 4 (H3K4me) correlates with active promoters and enhancers. These post-translational epigenetic modifications are deposited by Set1/Trithorax histone methyltransferase enzymes, which reside in large multi-protein complexes. While, MLL3 and MLL4 are predominantly associated with the deposition of H3K4me1 at enhancers, recent data in mouse embryonic stem cells indicate that MLL2 complex is implicated primarily in catalysing the deposition of the methyl mark at ‘bivalent’ developmentally regulated promoters. With respect to MLL1, genome-wide data shown its association with a subset of active promoters/enhancer. Importantly, the mechanisms governing the occupancy of the Set1/Trithorax complexes are still poorly understood. Moreover, as mentioned above, the function of the MLL complexes has been deeply studied in (mouse) embryonic stem cells, while their role in adult stem cells is still quite unclear.

In this manuscript, Goveas and coworkers investigated the role of MLL1 in the intestinal epithelium. By a clever ligand-induced conditional mutagenesis strategy, there were able to abrogate the expression of MLL1 in the gut epithelium of young mice. With this approach, the authors conclude that MLL1 is necessary for preserving the integrity of the at the base of the crypts where the intestinal stem cells (ISCs) are located. Moreover, it seems that MLL1 also implicated in maintaining the balance between secretory and absorptive cell lineages. Indeed, genetic deletion of MLL1 triggers the down-regulation of several genes, among those Jaml, a cell adhesion molecule. The effects of MLL1 deletion were further studied in organoids generated from isolated crypts.

Overall, the data presented here are robust and interesting. The paper is well-written and easy to read. The Achilles tendon for this paper is the lack of mechanism behind the effects observed upon genetic deletion of MLL1. Without further insights into the mechanism of MLL1 regulation of ISCs I would say that this manuscript does not provide a major advance in our knowledge of MLL1 or in its role in adult stem cells biology.

I thus would suggest few lines of experiments that can help the authors to elucidate MLL’s role in ISCs.

1) It is unclear whether the enzymatic activity of MLL1 is implicated in the observed phenotype. Moreover, the enzymatic role for Set1a has been recently put under discussion (PMID: 28939616). The authors should perform experiments in this direction: attempting the rescue the genetic deletion using enzymatically inactive MLL1, or by similar approaches. Furthermore, the analysis of ChIPseq data for H3K4me will also help in dissecting the methyltransferase-dependent and methyltransferase-independent functions of MLL1.

2) The authors should identify direct versus indirect effect of MLL1 deletion by combining RNAseq with ChIPseq data.

3) It has been reported that MLL proteins do not activate gene expression but rather prevents a default repressed state, often related to Polycomb occupancy. Is thus PRC1 and/or PRC2 accumulated at down regulated genes? Or is their silencing due to DNA methylation?

4) Since the targeting of MLL1 to specific loci is still poorly understood, the authors could further investigate this aspect by analysing the interactome of MLL1 in ISCs or in organoids.

**Have all data underlying the figures and results presented in the manuscript been provided?**

Reviewer #1: Yes

Reviewer #2: Yes

Reviewer #3: Yes

PLOS authors have the option to publish the peer review history of their article (what does this mean?). If published, this will include your full peer review and any attached files.

Reviewer #1: No

Reviewer #2: No

Reviewer #3: No

---

## [Decision Letter · Decision Letter 1]

24 Jun 2021

Dear Dr Kranz,

Thank you very much for submitting your Research Article entitled 'MLL1 is required for maintenance of intestinal stem cells and the expression of the cell adhesion molecule JAML' to PLOS Genetics.

The manuscript was fully evaluated at the editorial level and by independent peer reviewers. We have now assessed the reports from two of the original reviewers. Unfortunately, reviewer 1 was not available, but was able to suggest an alternative reviewer in their place. The new reviewer was overall positive about the paper and felt that many of the original concerns were fully addressed. However, this reviewer also raised an additional concern, that the expression changes of *Jaml* could be an artefact of inserting a cassette into the *Mll* gene (the reviewer also references the possibility of mouse strain specific issues as discussed in Mueller et al PMID: 32901003). They highlight the much larger downregulation of the *Jaml* gene compared to other loci as a point of concern, as well as the lack of H3K4me3 changes at *Jaml* (also a concern for reviewer 2). This possibility needs to be carefully considered and addressed in a carefully written response. Further analysis or additional data may be needed. Any other issues raised by the reviewers do not need to be considered in the response. Based on the reviews, we will not be able to accept this version of the manuscript, but we would be willing to review a revised version. We cannot, of course, promise publication at that time.

If you decide to revise the manuscript for further consideration at PLOS Genetics, please aim to resubmit within the next 60 days, unless it will take extra time to address the concerns of the reviewers, in which case we would appreciate an expected resubmission date by email to plosgenetics@plos.org.

[LINK]

We are sorry that we cannot be more positive about your manuscript at this stage. Please do not hesitate to contact us if you have any concerns or questions.

Yours sincerely,

Tom Milne

Guest Editor

PLOS Genetics

Wendy Bickmore

Section Editor: Epigenetics

PLOS Genetics

Reviewer's Responses to Questions

**Comments to the Authors:**

Reviewer #2: The revised manuscript further strengthened the major conclusion that Mll1 is required for intestinal stem cell maintenance and expression of certain key transcriptional regulators and JAML. The phenotypic and gene expression analyses are rigorous. The conclusions are significant and improve our understanding of the critical role of Mll1 in somatic stem cell biology.

The mechanisms by which Mll1 regulate these few genes are unknown. Mll1 ChIP is indeed technically challenging. The newly added H3K4me3 ChIP-qPCR data on the affected genes are helpful. However, it is somewhat disappointing that the authors did not pursue a more thorough assay to find out whether H3K4me3 at JAML is truly affected, since JAML is claimed to be the single most Mll1-dependent gene in these cells (and in the title!) and may be functionally relevant for ISC regulation by Mll1. It is surprising to see almost no H3K4me3 signal at the JAML promoter, since this gene seems to be expressed at a pretty high level and there is a well-established correlation of gene expression level and H3K4me3 level at transcription start site. The gene may have multiple TSSs and the choice could be tissue specific. In fact, H3K4me3 peak is probably the most reliable mark for an active true TSS. The primers most likely have missed the true TSS in this cell type. It matters because data presented in the current manner may (mis-)lead the readers to think that Mll1 must act on its most important target gene in intestinal stem cells through a methylation-independent way. H3K4me3 ChIP-seq is really not much more work than ChIP-qPCR, and would show the genome-wide impact of losing Mll1. As the H3K4 methylation-dependent and -independent activities of Mll1 remain poorly understood, the integrative analysis of the effects of Mll1 loss on genome-wide methylation and expression may provide some insight that could be interesting and useful to the epigenetics research field. Even if ChIP-seq is not done, the authors should at least test H3K4me3 by qPCR using primers at all alternative TSSs in updated database. There are also large number of H3K4me3 ChIP-seq data from mouse cells (especially ES cells) in databases that could aid in primer selection, though TSS in ISCs could be different. In fact, using publicly available mouse ES cell H3K4me3 ChIP-seq data show a prominent peak near the center of this gene (an alternative TSS), not at the annotated TSS detected by the primers used in this manuscript. Anyway, simply using one primer pair is not convincing (and could be misleading) for this target that is claimed to be the most important Mll1 target in ISCs.

Reviewer #3: In my report, I have kept at minimum my suggestions (only 4 points), in order to allow the authors to focus on those important aspects.

Yet, none of the criticisms I made were addressed.

The suggestions I made are important and should addressed before publication.

Reviewer #4: Goveas et al. have shown a requirement of the H3K4 methyltransferase Mll1(Kmt2a) for small intestine homeostasis, adding up a new stem cell compartment to the Mll1 repertoire. They have used an elegant conditional mouse mutagenesis strategy to delete Kmt2a in the whole body except HSCs (Rosa-CreER + WT BM transplantation) or specifically in the intestinal compartment (Villin-CreER). In both strategies they have observed a rapid decline of the mouse health with collapse of the intestinal function.

In this revised manuscript, the authors addressed successfully some of the reviewer’s concerns (note: I have not reviewed the first submission). However, there is a key aspect that has been missed and I believe needs to be clearly addressed:

Gene expression analysis of LGR5+ SCs did show modest fold changes in a small number of transcripts with only one transcript (Jaml) being significantly (34fold) down-regulated. Additional transcriptome analysis of P1 organoids has uncovered Mcam as the most down-regulated transcript.

Jaml (Amica1) and Mcam are neighbouring genes to Kmt2a. These 2 genes are by far the most down-regulated genes in RNAseq in Lgr5+_SC and P1 organoids. This is a potential source of artefacts since the transgenesis process has introduced a cassette that could disrupt expression of adjacent genes or even introduce mouse strain specific germline expression bias similar to this reference (Mueller, Lange et al. 2020). To strengthen this concern the authors have not identified many other significantly up- or down-regulated genes with more than 1fold change (log2) in ISCs. Moreover, Mll1 deletion did not cause a decrease on H3K4me on the Jaml promoter (Fig4H). The appropriate control mice would be a non-induced Mll1FC/FC to buffer any potential effects of transgenesis, in addition to Mll1FC/+ used by the authors. This is a key control since the authors base a lot of the discussion in the potential effects of Jaml down-regulation in the phenotype observed.

I understand that re-doing these experiments in the mice is time-consuming but they would be important to the wider community using this Mll1 mouse model in future experiments. Organoid cultures could be used as a quicker/cheaper alternative. Gene editing or RNAi on Mll1 should also be used to exclude any effects of the targeting allele/Cre recombination on Jaml expression (or other expressed neighbour genes such as Mcam).

**Have all data underlying the figures and results presented in the manuscript been provided?**

Reviewer #2: Yes

Reviewer #3: Yes

Reviewer #4: Yes

PLOS authors have the option to publish the peer review history of their article (what does this mean?). If published, this will include your full peer review and any attached files.

Reviewer #2: No

Reviewer #3: No

Reviewer #4: No

---

## [Decision Letter · Decision Letter 2]

30 Oct 2021

Dear Dr Kranz,

We are pleased to inform you that your manuscript entitled "MLL1 is required for maintenance of intestinal stem cells." has been editorially accepted for publication in PLOS Genetics. Congratulations!

Reviewer 4 has requested a potential explanation for the difference in phenotype between the mice and organoids, and if you have any speculation about the answer, it may be appropriate to add a few lines to the discussion upon final formatting.

Yours sincerely,

Tom Milne

Guest Editor

PLOS Genetics

Wendy Bickmore

Section Editor: Epigenetics

PLOS Genetics

Comments from the reviewers (if applicable):

Reviewer's Responses to Questions

**Comments to the Authors:**

Reviewer #2: This revision found an important error in the manuscript and explained the lack of H3K4me3 at Jaml “TSS”. The western data help consolidate the findings on other stem cell transcription factors. I appreciate the efforts the authors have taken to carefully examine this candidate gene. While the story leaves much to be studied about how MLL1 regulates stem cells at the mechanistic level, the current manuscript and the existing literature do provide a coherent picture on the phenotypic requirement of MLL1 in stem cell function. I think it can be accepted for publication.

Reviewer #4: The authors have properly addressed my concerns regarding Jaml expression by a careful re-analysis of the RNA-seq data. This led to removing Jaml as the potential link between Mll1-KO and the phenotype observed. However, the authors did not test a plausible functional explanation for their results other than the observed changes in RNA-seq for some transcription factors that could be associated with ISC function. Moreover, there is no explanation for the potential conflict between the phenotype in the mice (lower mitotic index, reduced stemness, more differentiation) and the organoids where you see a more cystic-like and immature state (less differentiation). Can the authors address this in their manuscript?

**Have all data underlying the figures and results presented in the manuscript been provided?**

Reviewer #2: None

Reviewer #4: Yes

PLOS authors have the option to publish the peer review history of their article (what does this mean?). If published, this will include your full peer review and any attached files.

Reviewer #2: **Yes: **Hao Jiang

Reviewer #4: No

**Data Deposition**

http://datadryad.org/submit?journalID=pgenetics&manu=PGENETICS-D-20-01516R2

**Press Queries**

---

## [Editor Report · Acceptance letter]

16 Nov 2021

PGENETICS-D-20-01516R2 

MLL1 is required for maintenance of intestinal stem cells. 

Dear Dr Kranz, 

We are pleased to inform you that your manuscript entitled "MLL1 is required for maintenance of intestinal stem cells." has been formally accepted for publication in PLOS Genetics! Your manuscript is now with our production department and you will be notified of the publication date in due course.

With kind regards,

Livia Horvath

PLOS Genetics

On behalf of:
